# Insights into cargo sorting by SNX32 and its role in neurite outgrowth

Jini Sugatha[1]*, Amulya Priya[2†], Prateek Raj[3†], Ebsy Jaimon[4†], Uma Swaminathan[5], Anju Jose[6], Thomas John Pucadyil[5], Sunando Datta[1]*

[1]Indian Institute of Science Education and Research, Bhopal, Bhopal, India; [2]SickKids Research Institute, Hospital for Sick Children, Toronto, Canada; [3]Molecular Biophysics Unit, Indian Institute of Science Bangalore, Bangalore, India; [4]Department of Biochemistry, Stanford University, Stanford, United States; [5]Indian Institute of Science Education and Research Pune, Pune, India; [6]Amala Cancer Research Centre, Thrissur, India

**Abstract** Sorting nexins (SNX) are a family of proteins containing the Phox homology domain, which shows a preferential endo-membrane association and regulates cargo sorting processes. Here, we established that SNX32, an SNX-BAR (Bin/Amphiphysin/Rvs) sub-family member associates with SNX4 via its BAR domain and the residues A226, Q259, E256, R366 of SNX32, and Y258, S448 of SNX4 that lie at the interface of these two SNX proteins mediate this association. SNX32, via its PX domain, interacts with the transferrin receptor (TfR) and Cation-Independent Mannose-6-Phosphate Receptor (CIMPR), and the conserved F131 in its PX domain is important in stabilizing these interactions. Silencing of SNX32 leads to a defect in intracellular trafficking of TfR and CIMPR. Further, using SILAC-based differential proteomics of the wild-type and the mutant SNX32, impaired in cargo binding, we identified Basigin (BSG), an immunoglobulin superfamily member, as a potential interactor of SNX32 in SHSY5Y cells. We then demonstrated that SNX32 binds to BSG through its PX domain and facilitates its trafficking to the cell surface. In neuroglial cell lines, silencing of SNX32 leads to defects in neuronal differentiation. Moreover, abrogation in lactate transport in the SNX32-depleted cells led us to propose that SNX32 may contribute to maintaining the neuroglial coordination via its role in BSG trafficking and the associated monocarboxylate transporter activity. Taken together, our study showed that SNX32 mediates the trafficking of specific cargo molecules along distinct pathways.

## Editor's evaluation

This manuscript presents a series of important findings about the roles of the BAR-domain containing protein SNX32 in endosomal cargo sorting and in neurite outgrowth. The authors provide compelling evidence for their claims, which will be of interest to those working not only in membrane trafficking but also to cell biologists in general with an interest in neurobiology.

## Introduction

Approximately 25% of the human genome encodes for integral membrane proteins (around 5500 proteins). Efficient intracellular transport of these proteins and associated proteins and lipids (together termed 'cargoes') is essential for organelle biogenesis, maintenance, and quality control. This is an extensive field that encompasses the secretory, endosomal, lysosomal, and autophagic pathways, all fundamental features of eukaryotic cells. Efficient integration of these pathways is essential for cellular

*For correspondence:
ssjini@outlook.com (JS);
sunando@iiserb.ac.in (SD)

†These authors contributed equally to this work

Competing interest: The authors declare that no competing interests exist.

organization and function, with errors leading to numerous diseases, including those associated with aging and neurodegeneration.

The endosomal network is composed of a series of distinct compartments that function to efficiently sort and transport cargo proteins between two distinct fates: either sorting for degradation in the lysosome or retrieval from this fate for recycling and transport to an array of organelles that include the cell surface, and the biosynthetic and autophagic pathway. The sorting nexin family is a conserved group of proteins with diverse roles in regulating the function of the endosomal network. Other than the PX-driven lipid specificity, much of the functional diversity displayed by the family could be apprehended by its ability to indulge in protein dimerization and oligomerizations (*Simonetti et al., 2019*). It is interesting to note that individual SNX-BAR proteins incapable of remodeling membrane in vitro (SNX5, SNX6, SNX7, SNX30, SNX32, etc.) are observed to undergo heterodimeric interactions with SNX-BAR proteins possessing an intrinsic membrane remodeling capacity (SNX1, SNX2, SNX4, SNX8, etc.) (*van Weering et al., 2012*). Protein–protein interactions thus add layers of intricate regulations, increasing the complexity of the system (*van Weering et al., 2012*).

Conserved throughout eukaryotes, defects in the function of these proteins are linked with various pathologies, including Alzheimer's disease (AD) (*Kim et al., 2017*), where they are associated with trafficking and processing (*Simonetti et al., 2019*; *Wang et al., 2014*; *Wang et al., 2013*; *van Weering et al., 2010*), Down's syndrome (DS) (*Vieira et al., 2021*), cancer (*Sharma et al., 2020*), schizophrenia, hypertension (*Huang et al., 2023*), thyroid disorders, and epilepsy (*Vieira et al., 2021*). The pathological implications of sorting nexin is an emerging area of extensive investigation (*Vieira et al., 2021*).

Sorting nexin 32 (SNX32), also known as SNX6b (*Simonetti et al., 2017*), on account of sequence similarity to its paralogue sorting nexin-6 (*van Weering et al., 2010*), is an underexplored member of the sorting nexin-containing BAR domain (SNX-BAR) sub-family of sorting nexins (*van Weering et al., 2010*). SNX32 is a component of the recently identified endosomal SNX-BAR sorting complex for promoting exit 1 (ESCPE-1), a heterodimer of SNX1/SNX2 associated with SNX5/SNX6/SNX32. ESCPE-1 regulates the sequence-dependent sorting of transmembrane cargo by recognizing and binding to a specific bipartite motif present within their cytosolic tail, an interaction mediated by the SNX5, SNX6, and SNX32 subunits (*Simonetti et al., 2019*; *Simonetti et al., 2017*; *Kvainickas et al., 2017*; *Wassmer et al., 2007*). The previous reports suggest that the ESCPE-1 complex identifies a bipartite signal sequence ($\Phi \times \Omega \times \Phi$ where $\Phi$ is hydrophobic and $\Omega$ is amino acid with aromatic side chains) present in the cytosolic tail of the prototypical cargoes and assists in its retrieval and recycling to TGN as well as to the surface (*Simonetti et al., 2019*). Even though it is referred that similar to its homologs SNX5/SNX6, SNX32 contributes to the trafficking of the CIMPR, it is yet to be demonstrated experimentally. Moreover, SNX32 is absent in non-higher metazoans, and in humans, its expression is higher in the brain compared to other tissues (http://www.gtexportal.org/home/gene/SNX32). This prompted us to investigate the role of sorting nexin in neuronal differentiation. In spite of heavy reliance on endocytic trafficking for functional fluidity in neurons, the underlying interlaced network of sorting leading to recycling or degradation remains elusive (*Lasiecka and Winckler, 2011*; *Horton and Ehlers, 2003*). In this study, we have investigated the interaction of SNX32 within the SNX family, its lipid affinity, and its role in the intracellular trafficking of CIMPR, transferrin receptor, and Basigin. We begin to define the function of SNX32 in intracellular trafficking in neuronal cell lines, revealing an important contribution to the process of neurite outgrowth.

## Results

### SNX32 undergoes BAR domain-assisted interaction with SNX1 and SNX4

The characteristic ability of the SNX-BAR family of proteins to undergo oligomerization directed us to examine the ability of SNX32 to participate in such interactions within the family. We utilized a GFP nanobody-based (*Fridy et al., 2014*; *Kubala et al., 2010*) immunoprecipitation (GBP-IP) assay where GST-GBP immobilized on glutathione sepharose beads was incubated with extracts from HEK293T cells overexpressing GFP/GFP-SNX1/GFP-SNX4/GFP-SNX8/GFP-SNX32 and HA-SNX32. The eluates were resolved through SDS-PAGE and subjected to immunoblotting. Immunoblotting using an anti-HA antibody showed that GFP-SNX1 (0.98), GFP-SNX4 (0.92), GFP-SNX8 (0.62), and SNX32 (0.85) efficiently precipitated HA-SNX32 (*Figure 1A and B*), which is also in agreement with the report by

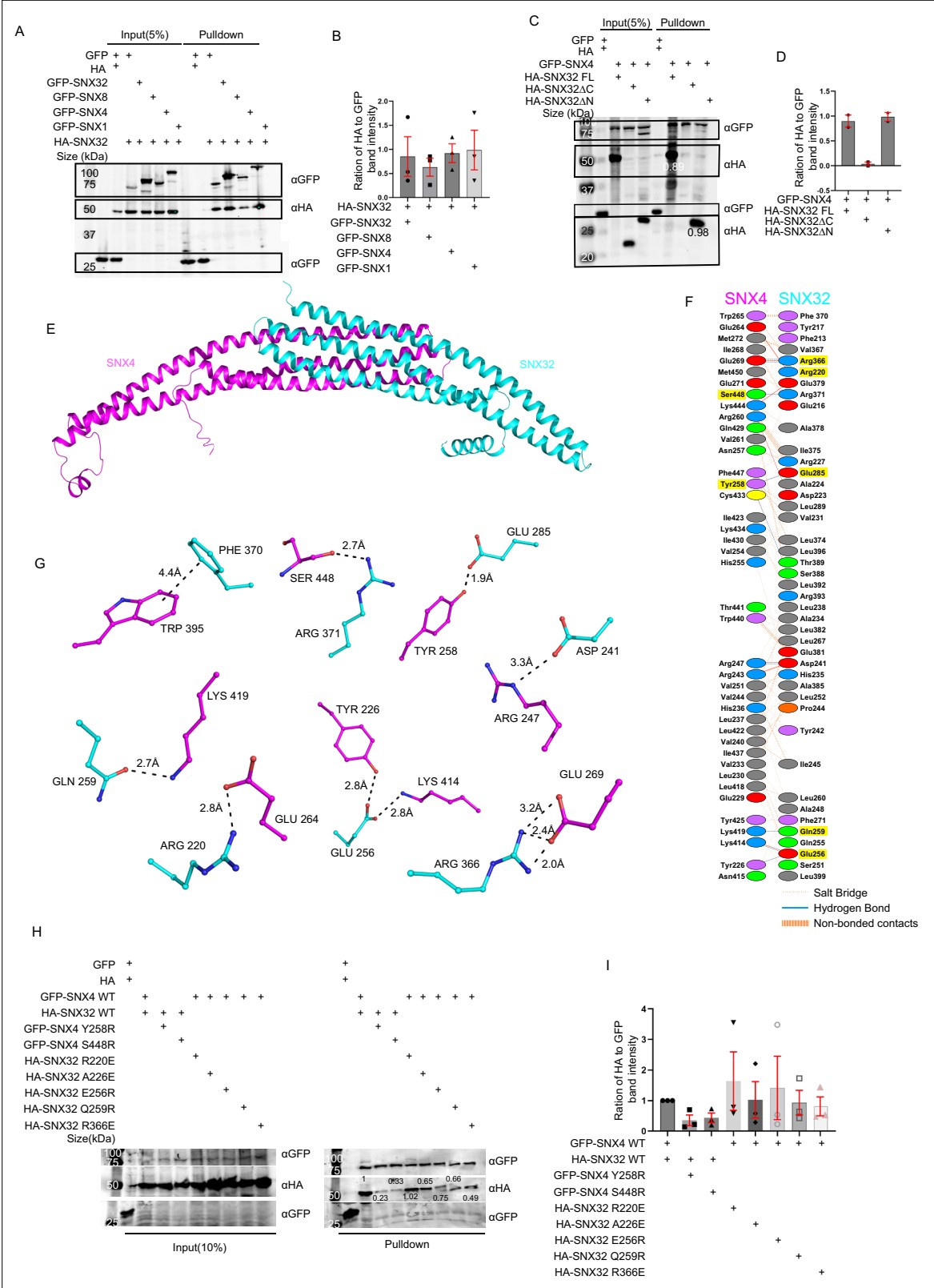

**Figure 1.** SNX32 undergoes BAR domain-mediated association with SNX4. (**A**) GBP co-immunoprecipitation of GFP/HA-tagged SNX-proteins transiently transfected in HEK293T cells showing GFP-SNX/GFP-SNX4/GFP-SNX8 or GFP-SNX32 efficiently precipitating HASNX32 (representative immunoblot out of three biological replicates, values represent the ratio of HA to GFP band intensity). (**B**) Plot representing quantifications of the three independent experiments of GBP co-immunoprecipitation of GFP/HA-tagged SNX-proteins represented in (**A**). (**C**) Co-immunoprecipitation of GFP/HA-

*Figure 1 continued on next page*

*Figure 1 continued*

tagged SNX-proteins transiently transfected in HEK293T cells showing GFP-SNX4 precipitating HA-SNX32ΔN (representative immunoblot out of three biological replicates, values represent the ratio of HA to GFP band intensity). (**D**) Plot representing quantifications of the three independent experiments of GBP co-immunoprecipitation of GFP/HA-tagged SNX-proteins represented in (**C**). (**E**) Homology model of BAR domains of SNX32-SNX4 complex. (**F**) Schematic depiction of the amino acid residues lining the heterodimeric interface of SNX32 (cyan)-SNX4 (magenta). (**G**) Polar interactions present at the dimeric interface of SNX32 (cyan)-SNX4 (magenta). (**H**) GBP co-immunoprecipitation of GFP/HA-tagged SNX-proteins transiently transfected in HeLa cells showing a difference in the amount of GFP-SNX4 precipitated HA-SNX32 mutants (representative immunoblot out of three biological replicates, values represent the ratio of HA to GFP band intensity). (**I**) Plot representing quantifications of the three independent experiments of GBP co-immunoprecipitation of GFP/HA-tagged SNX-proteins represented in (**H**).

The online version of this article includes the following source data and figure supplement(s) for figure 1:

**Source data 1.** GBP-IP of GFP/HA-tagged SNX-proteins transiently transfected in HEK293T cells showing GFP-SNX1/GFP-SNX4/GFP-SNX8 or GFP-SNX32 efficiently precipitating HA-SNX32.

**Source data 2.** Co-immunoprecipitation of GFP/HA-tagged SNX-proteins transiently transfected in HEK293T cells showing GFP-SNX4 precipitating HA-SNX32ΔN.

**Source data 3.** GBP co-immunoprecipitation of GFP/HA-tagged SNX-proteins transiently transfected in HeLa cells showing difference in the amount of GFP-SNX4 precipitated HA-SNX32 mutants.

**Figure supplement 1.** SNX32 undergoes BAR domain-mediated association with SNX1.

**Figure supplement 1—source data 1.** Co-immunoprecipitation of GFP/HA-tagged SNX-proteins transiently transfected in HEK293T cells showing the co-precipitation of GFP-SNX1 and HA-SNX32ΔN.

*Sierecki et al., 2014*. To identify the limiting region on SNX32 that contributes to these interactions, we generated two deletion constructs, an SNX32ΔN, spanning the BAR domain (*Sierecki et al., 2014*) (167–404), and SNX32ΔC, spanning the PX domain (*Chandra et al., 2019*) (1–166) (*Figure 1—figure supplement 1A*). The GBP-IP of GFP-SNX1/GFP-SNX4 and HA-SNX32 full-length (FL)/HA-SNX32ΔC/HA-SNX32ΔN showed that GFP-SNX4 (*Figure 1C and D*)/GFP-SNX1 (*Figure 1—figure supplement 1B and C*) precipitated HA-SNX32FL and HA-SNX32ΔN but not HA-SNX32ΔC. To assess whether the two SNXs are involved in direct interaction, we attempted to purify individual interacting partners from bacterial expression systems. Though His-SNX32ΔC was purifiable (*Figure 1—figure supplement 1D*), SNX32FL, SNX32ΔN, as well as the GST-SNX1/His-SNX32ΔN complex were unavailing due to the insolubility of the expressed proteins (*Figure 1—figure supplement 1E and F*).

## SNX32 and SNX4 interaction is mediated by the residues in the BAR domain interface of the SNXs

Our experimental data showed that SNX32 could interact with SNX1/SNX4 through its BAR domain. In order to identify the critical residues in maintaining the association between these SNXs, we predicted the structure of BAR domain heterodimers using the Alphafold2-Multimer (ColabFold) program (*Jumper et al., 2021*; *Mirdita et al., 2021*; *Evans et al., 2021*).

Generated models showed a high overall predicted local distance difference test (pLDDT) score, suggesting the high accuracy of the models. A few stretches (residues at the kinks between two helices) and several terminal residues have low pLDDT scores. However, most of the dimeric interface regions have a pLDDT score of more than 85. To assess the accuracy of our predictions, we modeled the homodimer of the SNX1-BAR domain for which the X-ray crystal structure (PDB ID: 4FZS) as well the cryo-EM structures (PDB ID: 7D6D and 7D6E) were already available. A comparison of the predicted model with the experimentally known SNX1-BAR homodimer revealed that the predicted model is highly consistent with the cryo-EM structure, thus resembling the membrane-bound SNX1 homodimer.

The predicted heterodimeric models of SNX32-BAR with SNX1-BAR and SNX4-BAR adopt a banana-shaped structure, a characteristic of conventional BAR domains (*Figure 1E–G*, *Figure 1—figure supplement 1G–L*). In the case of the SNX1-SNX32 BAR heterodimer (*Figure 1—figure supplement 1G–I*), the buried surface area (BSA) of SNX1-BAR and SNX32-BAR is 2572 Å$^2$ and 2519 Å$^2$, respectively. Similarly, the BSA of SNX4-BAR and SNX32-BAR in the SNX4-SNX32 BAR heterodimer (*Figure 1E–G*) is 2514 Å$^2$ and 2505 Å$^2$, respectively. High BSA represents the formation of strong heterodimers. On a closer look at the interphase of the SNX4-SNX32 complex, there are seven salt bridges and nine hydrogen bonds, whereas, in the case of the SNX1-SNX32 complex, there are four

salt bridges and six hydrogen bonds. The overlay of SNX1-SNX32 and SNX4-SNX32 revealed that the curvature attained in either case is comparable. The comparison of SNX1-SNX32 (*Figure 1—figure supplement 1G–I*) with SNX32-SNX32 (*Figure 1—figure supplement 1J–L*) revealed that Y242, R366, and K400 (*Figure 1—figure supplement 1H and K*) of SNX32 participate in stabilizing both homodimeric and the heterodimeric interface, whereas D223, L399, and E402 (*Figure 1—figure supplement 1H and I*) uniquely mediate the interactions at the heterodimeric interface. Similarly, in the case of SNX4-SNX32, R220, E256, and R366 of SNX32 (*Figure 1G*) participate in both hetero-/homodimeric interfaces, whereas D241, Q259, E285, F370, and R371 (*Figure 1G and F*) participate only in the heterodimeric interface. Of note, F370 is involved in mediating a pi–pi interaction with F264 of SNX32 and W265 of SNX4 (*Figure 1G and F*). Overall, the heterodimeric interface of SNX1-SNX32 and SNX4-SNX32 displays extensive hydrophobic interactions along with several polar interactions.

Based on the above model, we next assessed whether the interaction between SNX32 and SNX4 could be disrupted by mutating the amino acids engaged in interactions at the interphase. Single mutations of S448R and Y258E in the GFP-SNX4 BAR domain and single mutations of R220E, E256R, Q259R, and R366E in the HA-SNX32 were carried out utilizing site-directed mutagenesis. Additionally, it was previously reported that SNX5S226E mutation disrupts its heterodimeric interaction with SNX1, whereas SNX5S226A does not (*Simonetti et al., 2019*; *Itai et al., 2018*); based on that, we looked into the structure of SNX32, which had an alanine in the 226th position and decided to include A226E in our study.

We performed GBP-IP using wild-type (WT) vs. mutants of GFP-SNX4/HA-SNX32. We observed that the HA-SNX32 WT was efficiently immuno-precipitated by GFP-SNX4 WT but not GFP-SNX4 S448R (0.23)/Y258E (0.33) (*Figure 1H and I*). Similarly, GFP-SNX4 WT immuno-precipitated the HA-SNX32 R220E (1.02), but the amount of precipitant of HA-SNX32 A226E (0.65), HA-SNX32 Q259R (0.75), HA-SNX32 E256R (0.66), and HA-SNX32 R366E (0.49) was comparably less (*Figure 1I*).

To further understand the in cellulo implications of the interaction of SNX32 with SNX1/SNX4, we carried out various biochemical and immunofluorescence-based studies.

## Subcellular localization of SNX32: The contribution of individual domains

We began with the study of subcellular localization of SNX32 in HeLa cells. As shown in *Figure 2—figure supplement 1A*, the expression of SNX32 in HeLa cells, which is an established system for studying intracellular trafficking, was comparable with that in human brain-derived cells such as U87MG. We generated a HeLa cell line stably expressing GFP-SNX32 from an inducible promoter of pLVX TRE3G. Upon treatment with a concentration of 500 ng/ml of doxycycline for 13 hr, the subcellular localization of SNX32 was punctate. Further, the cells were fixed, immunostained using an early endosomal marker (EEA1), Trans Golgi marker (TGN46), and juxtanuclear recycling endosomal marker (mCherry-Rab11), and images were acquired by a laser-scanning confocal microscope. As evident from the object-based colocalization calculated using Motion tracking (*Kalaidzidis, 2007*; *Rink et al., 2005*), GFP-SNX32 colocalized with early endosomal marker EEA1 (*Figure 2A and B*), which is also in accord with the study reported by *Simonetti et al., 2017*. We also observed that a notable amount of SNX32 localizes to the cell surface (*Figure 2C*). In addition, GFP-SNX32 partially colocalized with mCherry-Rab11 (*Figure 2D and B*) and TGN46 (*Figure 2B and E*), but less with GM130 (*Figure 2—figure supplement 1B*, *Source data 1*). The partial overlaps were reconfirmed in live-cell microscopy experiments (*Video 1*) as well as in super-resolution imaging (*Figure 2F and G*, *Videos 2 and 3*).

Confocal images revealed that HA-SNX32ΔN showed punctate localization scattered through the cytoplasm and did not show colocalization with known endosomal markers (*Figure 2—figure supplement 1C*). However, when co-expressed with the interacting partners, GFP-SNX4/GFP-SNX1, they localized on EEA1-positive compartments (*Figure 2—figure supplement 1D and E*). Distinctively, HA-SNX32ΔC was seen as a perinuclear cluster that colocalized with PI (4)P binding GFP-PH $^{OSBP}$ (*Figure 2—figure supplement 1F and G*). This prompted us to understand the lipid-binding specificity of each domain in cellulo.

We utilized phosphatidylinositol (PI) kinases-inhibitors phenyl arsine oxide (PAO) (*Shin et al., 2020*), a PI4 kinase inhibitor, or Wortmannin, a PI3 kinase inhibitor (*Powis et al., 1994*), to acutely deplete the intracellular PI(4)P or PI(3)P, respectively (*Niu et al., 2013*). To confirm the specificity of PAO treatment, its effect on GFP-PH$^{OSBP}$, a family of tethering proteins that bind specifically to PI(4)P (*Mesmin*

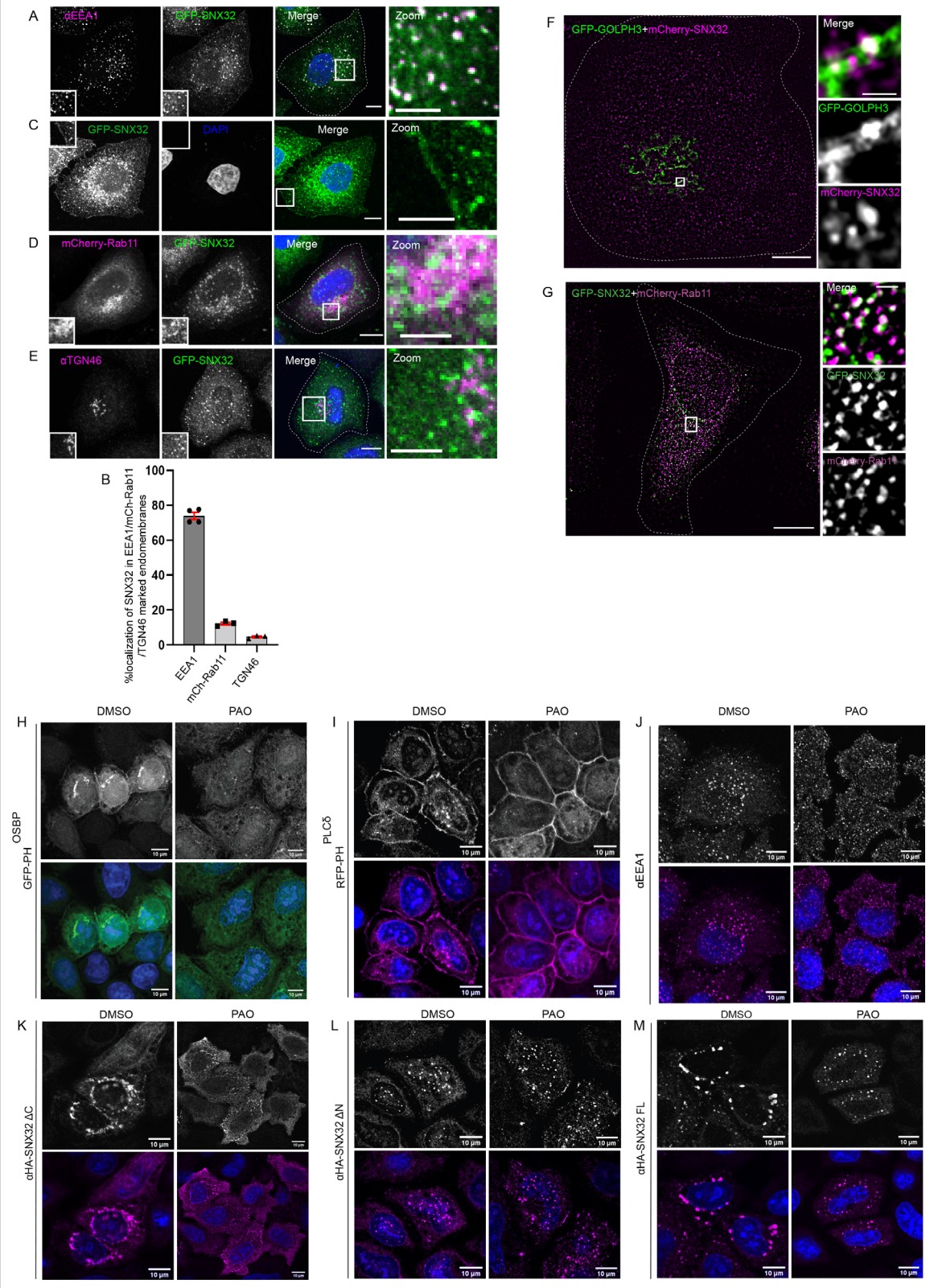

**Figure 2.** SNX32 undergoes PX domain-assisted localization to PI(4)P-enriched endosomal membranes in addition to early endosomes. (**A**) GFP-SNX32 localization with early endosomal marker EEA1. (**B**) Quantifications showing percentage colocalization of GFP-SNX32 in HeLa cells, data represent mean ± SEM (N = 3, n ≥ 60 cells per independent experiments). (**C**) GFP-SNX32 localization at cell periphery (**D**) GFP-SNX32 colocalization with perinuclear recycling endosomal marker mCherry-Rab11, (**E**) Colocalization of GFP-SNX32 with Trans Golgi Network marker TGN46. Scale bar 10 μm, inset 5 μm

*Figure 2 continued on next page*

*Figure 2 continued*

(magnified regions are shown as insets, inset of the merge is magnified and represented as zoom). (**F, G**) SIM image showing colocalization of (**F**) GFP-GOLPH3 and mCherry-SNX32. (**G**) GFP-SNX32 and mCherry-Rab11. Scale bar 10 μm, inset 1 μm (magnified regions are shown as insets). (**H–M**) PAO/DMSO treatment in HeLa cells overexpressing (**H**) GFPPH$^{OSBP}$, (**I**) RFP-PH$^{PLCδ}$, (**J**) endogenous EEA1, (**K**) HA-SNX32ΔC, (**L**) HA-SNX32ΔN, and (**M**) HA-SNX32FL. Scale bar 10 μm.

The online version of this article includes the following source data and figure supplement(s) for figure 2:

**Figure supplement 1.** BAR domain of SNX32 undergoes protein–protein interactions and assist in the early endosomal localization of SNX32.

**Figure supplement 1—source data 1.** Membrane-cytosol fractions of HeLa cells transiently transfected with HA-SNX32ΔC or GFP-PH$^{OSBP}$ showing PAO treatment causes delocalization of HA-SNX32ΔC and GFP-PH$^{OSBP}$ proteins to the cytosolic fraction, S-cytosol, P-membrane fraction.

**Figure supplement 2.** Lipid affinity of SNX32ΔN.

**Figure supplement 2—source data 1.** PIP strip membrane immunoblotted using His antibody showing preferential binding of His-SNX32ΔC to PI(3)P, PI(4)P, PI(5)P, PA.

**Figure supplement 2—source data 2.** PLiMAP images showing in-gel fluorescence of BDPE-crosslinked (top) and Coomassie blue stained (bottom), His-SNX32ΔC, His-SNX12, and GST-2xP4M domain of SidM after PLiMAP.

---

*et al., 2013*), was assessed. Similarly for confirming the specificity of Wortmannin treatment, its effect on EEA1, an FYVE domain-containing protein that specifically binds to PI(3)P (*Stenmark et al., 2002*), was examined. To negate any off-target effect RFP-PH$^{PLCδ}$, a phosphatidylinositol 4,5-bisphosphate (PI(4,5)P2) binding protein (*Garcia et al., 1995*) motif was used as a negative control. The cells un-transfected or overexpressing GFP-PH$^{OSBP}$, RFP-PH$^{PLCδ}$, GFP-SNX32, HA-SNX32ΔC, HA-SNX32ΔN were treated with PAO (15 μM)/ Wortmannin (200 nM) in uptake medium for 15 min at 37°C and proceeded for immunofluorescence. Consistent with the previous reports (*Niu et al., 2013*), PAO treatment caused redistribution of GFP-PH$^{OSBP}$ (*Figure 2H*), whereas Wortmannin treatment caused the redistribution of endogenous EEA1 (*Figure 2—figure supplement 1H*) with less or no effect on the distribution of other PIns binding markers (*Niu et al., 2013*), indicating the depletions were specific on either case (*Figure 2I and J*, *Figure 2—figure supplement 1I and J*). We observed that the PAO treatment caused redistribution of HA-SNX32ΔC from the perinuclear cluster to the cytosol (*Figure 2K*) with less or no effect on the localization of HA-SNX32ΔN

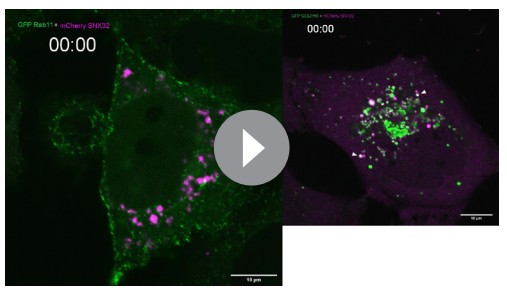

**Video 1.** On the left: live-cell imaging showing transient contact between SNX32 vesicles and Rab11-labeled recycling endosomes. HeLa cells co-transfected with plasmids encoding mCherry-SNX32 (magenta) and GFP-Rab11 (green). Videos were captured in free run mode, without intervals in Olympus FV3000 confocal laser-scanning microscope at 37°C,5% CO$_2$ with moisture control. ZDC-Z Drift compensation was used to correct focus drift during time courses. Frames were collected every 6.44 s for 9 min 58 s. Playback rate is 3 frames/s. The transient contact events are indicated by white arrowheads. On the right: live-cell imaging showing transient contact between SNX32 vesicles and GOLPH3-labeled Golgi compartment: HeLa cells stably expressing GFP-GOLPH3 (green) were transfected with plasmid encoding mCherry-SNX32 (magenta). Videos were captured in free run mode, without intervals in Olympus FV3000 confocal laser-scanning microscope at 37°C,5% CO$_2$ with moisture control. ZDC-Z compensation was used to correct focus drift during time courses. Frames were collected every 12.24 s for 9 min 59 s. The transient contact events are indicated by white arrowheads.

https://elifesciences.org/articles/84396/figures#video1

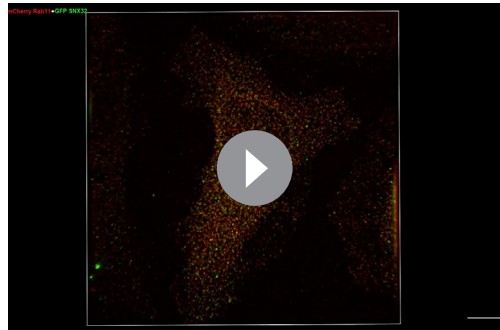

**Video 2.** SNX32 colocalize with Rab11-positive recycling endosomes. 3D-reconstructed SIM movie (Nikon N-SIM S) of HeLa cell showing colocalization of mCherry-Rab11-positive recycling endosomes (red) and GFP-SNX32 (green). A total of 26 Z planes of 0.25 μm step size was captured. For each Z plane and each wavelength, 15 images were captured (three different angle and five different phases) Scale bar 10 μm.

https://elifesciences.org/articles/84396/figures#video2

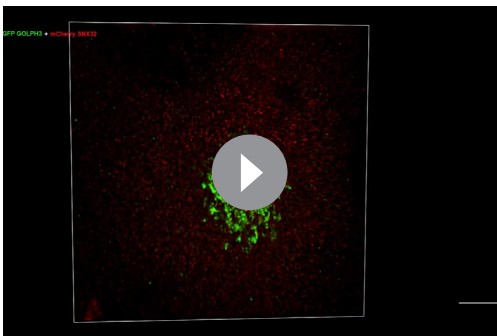

**Video 3.** SNX32 vesicles colocalize with GOLPH3-positive Golgi compartment. 3D-reconstructed SIM movie (Nikon N-SIM S) of HeLa cell showing colocalization of GFP-GOLPH 3 (green)-positive Golgi compartment and mCherry-SNX32 (red). A total of 28 Z planes of 0.30 μm step size was captured. For each Z plane and each wavelength, 15 images were captured (three different angles and five different phases),Scale bar 10 μm.

https://elifesciences.org/articles/84396/figures#video3

(*Figure 2L*). In contrast, Wortmannin treatment caused HA-SNX32ΔN to relocalize from the punctae to the cytosol (*Figure 2—figure supplement 1K*) with less effect on HA-SNX32ΔC (*Figure 2—figure supplement 1L*). For both the inhibitor treatments, the altered localization of the SNX was less pronounced in the case of the full-length protein (*Figure 2M*, *Figure 2—figure supplement 1M*). On the contrary, the effect of Wortmannin was much more pronounced in the localization of GFP-SNX1 and GFP-SNX4 (*Figure 2—figure supplement 1N and O*). The effect of PAO on HA-SNX32ΔC was further confirmed using a membrane fractionation experiment (*Seaman et al., 2009*). HeLa cells overexpressing GFP-PH^OSBP or HA-SNX32ΔC were incubated with PAO for 15 min at 37°C followed by snap-freezing to fracture the membrane and release the cellular contents. We found that the distribution of HA-SNX32ΔC was increased in the cytosolic fraction when treated with PAO (*Figure 2—figure supplement 1P*).

Hence, we hypothesized that the PIns affinity of the SNX32 PX domain might contribute to the SNX's association with the endomembrane. We employed protein-lipid overlay (*Dowler et al., 2002*) assay which enables a quantitative understanding of the lipid-interacting preference of lipid-binding proteins. The immunoblot of the PIP strip showed that the His-SNX32ΔC was bound to PI(3)P, PI(5)P, PI(4)P, and PA (*Figure 2—figure supplement 2A*). We further tested the binding specificity of the PX domain of SNX32 for PI(3)P and PI(4)P using Proximity-based Labelling of Membrane-Associated Proteins (PLiMAP) (*Jose et al., 2020*; *Jose and Pucadyil, 2020*). We did not observe any detectable binding of His-SNX32 ΔC to either PI(3)P or PI(4)P compared to SNX12 (*Priya et al., 2017*; *Pons et al., 2012*) or 2xP4M domain of SidM (*Brombacher et al., 2009*; *Figure 2—figure supplement 2B*).

## SNX32 regulates CIMPR trafficking

Further, to define the functional significance of the interaction between SNX32 and the SNX1, the cargo trafficking of CIMPR was investigated. It was previously reported that Cation-Independent Mannose-6-Phosphate Receptor (CIMPR) interacts with SNX32 (*Simonetti et al., 2019*) and the interaction follows the canonical sequence-dependent recognition similar to CIMPR and SNX5/SNX6 (*Simonetti et al., 2017*). Even though CIMPR is shown to be a potential cargo for SNX32 (*Simonetti et al., 2019*; *Simonetti et al., 2017*), the involvement of SNX32 in the trafficking of CIMPR was not yet reported; additionally, it has been shown that the suppression of SNX1 leads to the degradation of CIMPR (*Carlton et al., 2004*). Our results from confocal microscopy showed that SNX32 and SNX1 colocalize (*Figure 3A–D*) and co-transports CIMPR (*Video 4*). Further, we observed that the SNX1 knockdown (KD)/SNX6KD/SNX32KD (*Figure 3—figure supplement 1A*) resulted in the steady-state redistribution of CIMPR (*Figure 3E–H*), which was also seen to be correspondingly more with EEA1-positive early endosomes (*Figure 3I*). Moreover, the double KD of SNX1 and SNX32 did not show any significant difference in the redistribution of CIMPR (*Figure 3J and I*). On the contrary, the downregulation of SNX6 along with SNX32 showed a significant increase in the redistributed CIMPR (*Figure 3K and I*). Moreover, the effect of SNX32KD in redistribution of CIMPR was reaffirmed using shRNA-mediated SNX32 downregulation (*Figure 3—figure supplement 1B–F*). We did not find any significant reduction in the total level of CIMPR upon suppression of SNX6/SNX32 (*Figure 3—figure supplement 1G and H*). To further confirm whether the steady-state redistribution of CIMPR upon SNX32KD resulted from a defect in the kinetics of transport, we performed an antibody uptake assay utilizing CD8-CIMPR reporter endocytosis (*Seaman, 2004*). In control cells, after incubating the cells with an anti-CD8 antibody to label the cell surface CD8-CIMPR, the kinetics of transport revealed

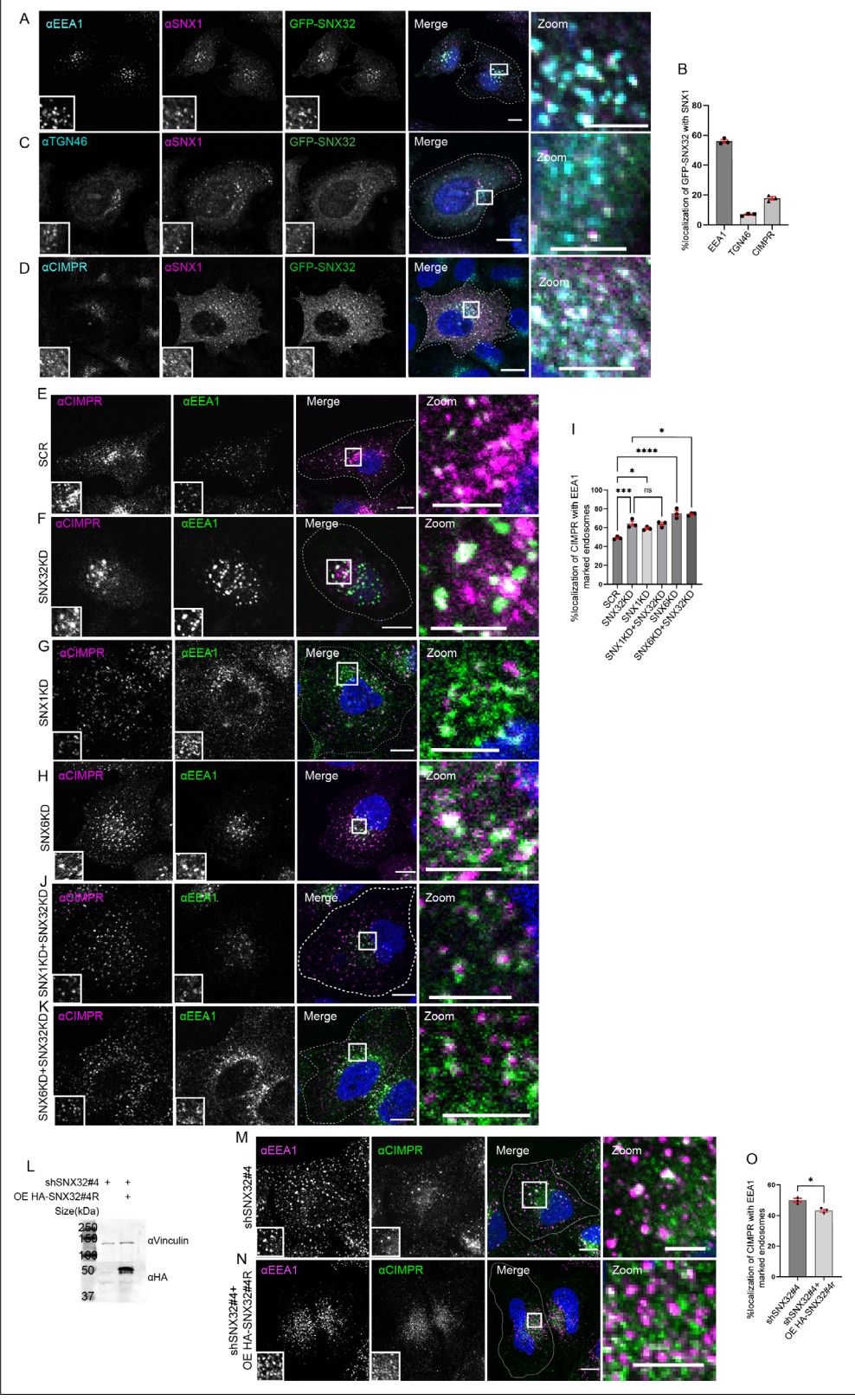

**Figure 3.** SNX32 interacts with and regulate the trafficking of CIMPR in HeLa cells. (**A–D**) SNX32 and SNX1
colocalizes with (**A**) early endosome marker EEA1. (**B**) Percentage localization of SNX32-SNX1 heterodimer
with EEA1, TGN46, and CIMPR, data represent mean ± SEM (N = 3, n ≥ 60 cells per independent experiments).
(**C**) Trans Golgi Network marker TGN46. (**D**) CIMPR. Scale bar 10 μm, inset 5 μm (magnified regions are shown

*Figure 3 continued on next page*

*Figure 3 continued*

as insets, inset of the merge is magnified and represented as zoom), (**E–K**) Followed by SMART pool-mediated gene downregulation of (**E**) scrambled (SCR), (**F**) SNX32, (**G**) SNX1, and,(**H**) SNX6, HeLa cells were immunostained using CIMPR, EEA1 and nuclei were counterstained with DAPI (blue). Scale 10 µm, inset 5 µm (magnified regions are shown as insets, inset of the merge is magnified and represented as zoom). (**I**) Quantification of percentage localization of CIMPR with EEA1, data represent mean ± SEM (N = 3, n ≥ 60 cells per independent experiments), p-value<0.0001 (***p<0.001, ****p<0.0001, *p<0.05), one-way ANOVA, Šídák's multiple-comparisons test. (**J, K**) Followed by SMART pool-mediated gene downregulation of (**J**) SNX32 and SNX1, (**K**) SNX32 and SNX6, HeLa cells were immunostained using CIMPR, EEA1 and nuclei were counterstained with DAPI (blue). Scale bar 10 µm, inset 5 µm (magnified regions are shown as insets, inset of the merge is magnified and represented as zoom). (**L**) Followed by shSNX32#4-mediated gene downregulation of SNX32 HASNX32#4r was overexpressed for 12 hr, immunoblotting was done using HA or vinculin antibody (representative immunoblot out of three biological replicates). (**M**) Followed by shRNA-mediated gene downregulation of SNX32 utilizing shRNA clone – shSNX32#4, HeLa cells were immunostained using CIMPR, EEA1 and nuclei were counterstained with DAPI (blue). (**N**) Followed by shSNX32#4-mediated gene downregulation of SNX32 HA-SNX32#4r was overexpressed for 12 hr, HeLa cells were immunostained using CIMPR, EEA1 and nuclei were counterstained with DAPI (blue). Scale bar 10 µm, inset 5 µm (magnified regions are shown as insets, inset of the merge is magnified and represented as zoom). (**O**) Quantification of percentage localization of CIMPR with EEA1, data represent mean ± SEM (N = 3, n ≥ 60 cells per independent experiments), p-value 0.0121 (*p<0.05), unpaired *t* test two-tailed.

The online version of this article includes the following source data and figure supplement(s) for figure 3:

**Figure supplement 1.** SNX32KD or SNX6 KD does not alter the total CIMPR protein in cells.

**Figure supplement 1—source data 1.** HeLa cells were transfected with Scramble/SNX32/SNX6 siRNA or SHC002, shSNX32#4, shSNX32#6 shRNA followed by cycloheximide treatment of 10 µg/ml for 6 hr.

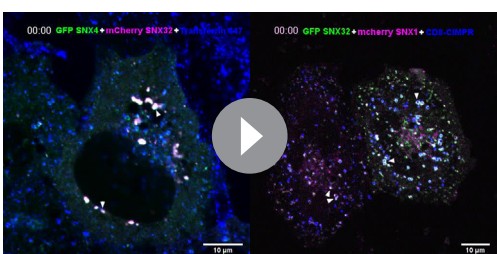

**Video 4.** On the left: SNX32, SNX4 co-traffic transferrin. HeLa cells co-transfected with plasmids encoding mCherry-SNX32 (magenta) and GFP-SNX4 (green) were allowed to uptake Alexa 647-labeled transferrin (blue) for 2 min. Videos were captured in free run mode, without intervals in Olympus FV3000 confocal laser-scanning microscope at 37°C,5% $CO_2$ with moisture control. ZDC-Z Drift compensation was used to correct focus drift during time courses. Frames were collected every 9.58 s for 4 min 47 s. Playback rate is 2 frames/s. The co-trafficking events are indicated by white arrowheads.On the right: SNX32, SNX1 co-traffic CD8-CIMPR: HeLa cells co-transfected with plasmids encoding mCherry-SNX1 (magenta) and GFP-SNX32 (green) and CD8-CIMPR (blue) were processed as detailed in the 'Materials and methods' section. Videos were captured in free run mode, without intervals in Olympus FV3000 confocal laser-scanning microscope at 37°C,5% $CO_2$ with moisture control. ZDC-Z Drift compensation was used to correct focus drift during time courses. Frames were collected every 6.520 s for 4 min 49 s. Playback rate is 3 frames/s. The co-trafficking events are indicated by white arrowheads. https://elifesciences.org/articles/84396/figures#video4

that the CD8-CI-MPR had reached Golgi after an incubation period of 30 min (*Figure 3—figure supplement 1I and J*). In contrast, the HeLa cells suppressed for SNX32, the internalized anti-CD8 antibody was observed to be accumulated in endosomes and were unable to reach Golgi after 30 min of chase period (*Figure 3—figure supplement 1J and K*). We further validated the effect of SNX32 downregulation on the early endosomal redistribution of CIMPR by compensating for the loss of SNX32. We observed that by overexpressing the shRNA-resistant SNX32(HA-shSNX32#4r), in the background of shSNX32#4-mediated SNX32 knockdown (*Figure 3L*), the colocalization of CIMPR with EEA1 was reduced (*Figure 3M–O*).

## SNX32 regulates transferrin trafficking

In addition to SNX1, SNX32 could also associate with SNX4 (*Figure 1A*). Previously, it has been reported that SNX4 suppression alters the levels of transferrin receptor (TfR) (*Traer et al., 2007*) by diverting it from the recycling route to the lysosomal degradative pathway. Though SNX32 is largely colocalized with SNX4 on EEA1-positive early endosomal compartments (*Figure 4A*), we also observed a small but significant SNX32 population on the transferrin receptor-positive recycling compartment (*Figure 4B*). Our results from confocal microscopy on paraformaldehyde (PFA)-fixed cells and live-cell video microscopy revealed

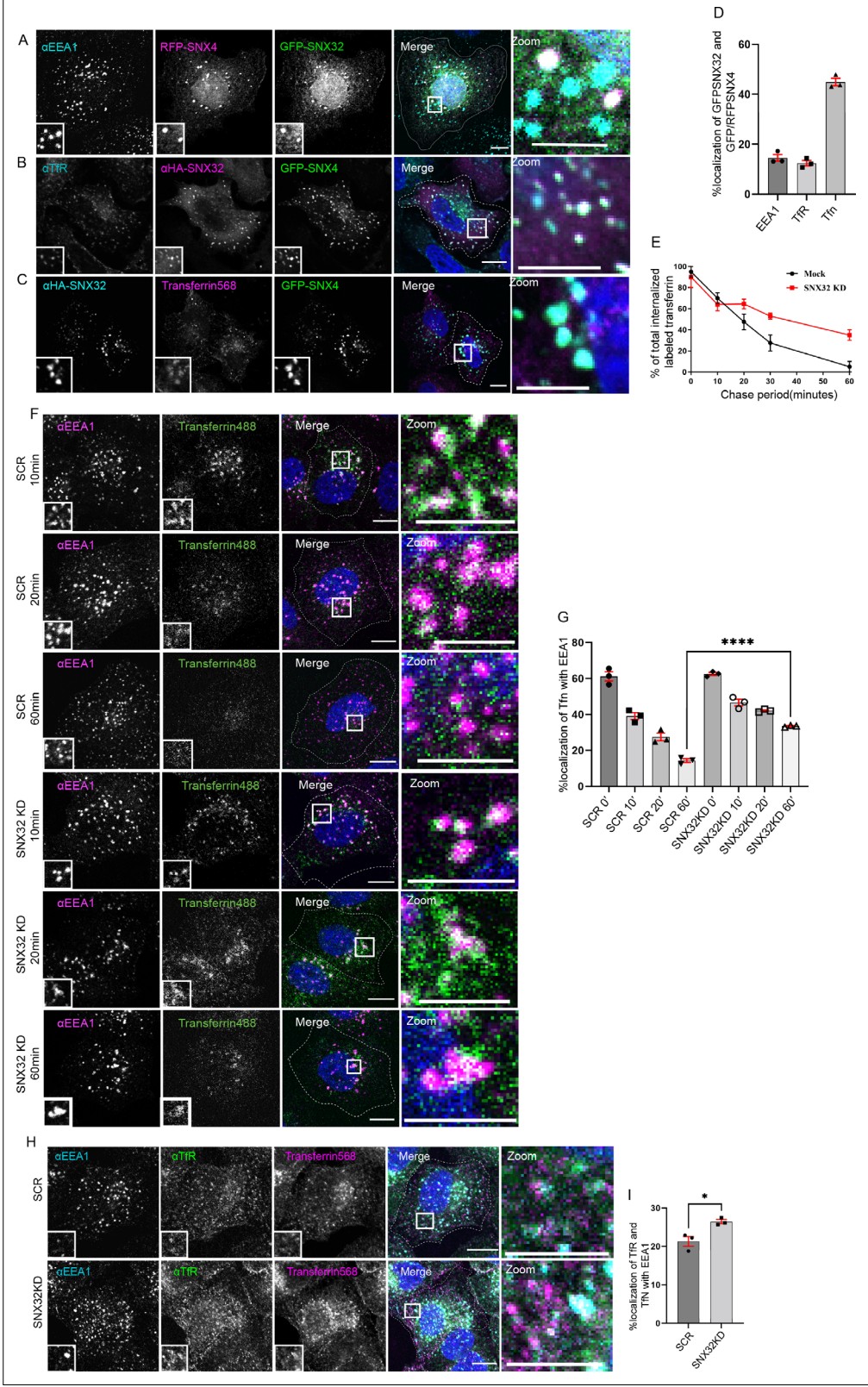

**Figure 4.** SNX32 interacts with and regulate the trafficking of transferrin-bound transferrin receptor in HeLa cells. (A–D) SNX32 and SNX4 colocalize with (A) early endosome marker EEA1 and (B) perinuclear recycling endosomal marker TfR. (C) Alexa Fluor 568 conjugated transferrin. Scale bar 10 µm, inset 5 µm (magnified regions are shown as insets, inset of the merge is magnified and represented as zoom). (D) Percentage localization of SNX32 and SNX4

*Figure 4 continued on next page*

*Figure 4 continued*

with EEA1, TfR, and transferrin, data represent mean ± SEM (N = 3, n ≥ 60 cells per independent experiments). (**E**) Quantitative estimation of the percentage of total internalized transferrin (Alexa Fluor 488 conjugated) in different chase periods measured by its total intensity of fluorescence in mock and SNX32-suppressed cells, data represent mean ± SEM (N = 3, n ≥ 60 cells per independent experiments). (**F**) Followed by SMART pool-mediated gene downregulation of scrambled (SCR) or SNX32, transferrin (Alexa Fluor 488 conjugated) pulse-chase experiment was carried out as described in the 'Materials and methods' section, the cells were fixed at specified timepoints, immunostained using early endosomal marker EEA1, and DAPI was used to stain nucleus. Scale bar 10 µm, inset 5 µm (magnified regions are shown as insets, inset of the merge is magnified and represented as zoom). (**G**) Quantification of percentage localization of transferrin (Alexa Fluor 488 conjugated) with EEA1 at corresponding timepoints, data represent mean ± SEM (N = 3, n ≥ 60 cells per independent experiments), p-value<0.0001 unpaired *t* test two-tailed (****p<0.0001). (**H**) Followed by SMART pool-mediated gene downregulation of scrambled (SCR) or SNX32, HeLa cells were allowed to uptake transferrin (Alexa Fluor 568 conjugated) for 30 min, 37°C, fixed and immunostained using EEA1, TfR, and DAPI was used to stain nucleus. Scale bar 10 µm, inset 5 µm (magnified regions are shown as insets, inset of the merge is magnified and represented as zoom). (**I**) Quantification of percentage localization of TfR with internalized transferrin and EEA1, data represent mean ± SEM (N = 3, n ≥ 60 cells per independent experiments), p-value 0.0206 (*p<0.05), unpaired *t* test two-tailed.

The online version of this article includes the following source data and figure supplement(s) for figure 4:

**Figure supplement 1.** Transferrin stuck at EEA1 due to loss of SNX32 could be rescued by overexpressing shRNA-resistant SNX32.

**Figure supplement 2.** SNX6 and SNX32 show an overlapping role in regulating the trafficking of transferrin in HeLa cells.

**Figure supplement 2—source data 1.** HeLa cells were transfected with scramble/SNX32/SNX6 siRNA or shc002, shSNX32#4, shSNX32#6 shRNA followed by cycloheximide treatment of 10 µg/ml for 6 hr, immunoblotting was done using TfR or vinculin antibody.

that SNX32 and SNX4 colocalize (*Figure 4C and D*) and co-transport with transferrin (*Video 4*). We then assessed the recycling of transferrin in HeLa cells depleted for SNX32 using siRNA (*Figure 3—figure supplement 1A*) or shRNA clones (*Figure 3—figure supplement 1B*). We observed no significant difference in the integral intensity of transferrin in the initial timepoints (0–10 min) of the chase period. However, in the course of a 60 min chase, the scrambled cells (SCR) followed a trend of diminishing integral fluorescence intensity (*Figure 4E*), whereas SNX32KD cells retained the intensity (*Figure 4E*). We characterized the nature of the endosomes in which transferrin was present and found that it was retained in the early endosomes positive for EEA1 (*Figure 4F and G*). Moreover, the receptor's distribution along with the transferrin revealed both being present in the EEA1-positive vesicles (*Figure 4H and I*). In agreement with this, the shRNA-mediated SNX32 downregulated (shSNX32) conditions (*Figure 3—figure supplement 1B*, *Figure 4—figure supplement 1A and B*) also showed similar effects. We further validated the effect of SNX32 downregulation in the early endosomal redistribution of transferrin by compensating for the loss of SNX32. We observed that by overexpressing the shRNA-resistant SNX32(HA-shSNX32#4r), in the background of shSNX32#4-mediated SNX32 knockdown, the colocalization of transferrin (Alexa Fluor 568 conjugated) with EEA1 was reduced (*Figure 4—figure supplement 1C and D*); however, the overexpression of the individual domains (HA-SNX32ΔN#4r, *Figure 4—figure supplement 1D and E*), HA-SNX32ΔC (*Figure 4—figure supplement 1D and F*) failed to show any such effect, suggesting their role in cargo trafficking (*Figure 4—figure supplement 1C–F*).

Since SNX6 is the closest homolog of SNX32, we investigated whether SNX6 downregulation (SNX6KD) also affects transferrin trafficking. Comparable to SNX32KD, SNX6KD also displayed that the transferrin was retained at EEA1-positive early endosome even after 30 min of chase period (*Figure 4—figure supplement 2A and B*). We then asked whether SNX32 and SNX6 have any overlapping roles in transferrin trafficking. To address this, we overexpressed GFP-SNX32 in SNX6KD cells and measured the transferrin colocalizing with the EEA1-positive compartment. As observed in *Figure 4—figure supplement 2C and D*, the colocalization of transferrin with EEA1 is significantly reduced in the SNX6KD cells expressing GFP-SNX32. We also observed that the downregulation of SNX32 or SNX6 did not alter the total TfR levels in HeLa cells (*Figure 4—figure supplement 2E and F*).

## Interplay of SNX32 with SNX4 and Rab11 in transferrin trafficking

Next, to have a better understanding of the SNX32-regulated transferrin trafficking pathways, we investigated transferrin trafficking with respect to the already established key regulators such as SNX4 or Rab11. As reported earlier (*Traer et al., 2007*), the downregulation of SNX4 resulted in increased colocalization of transferrin in the EEA1-positive compartment (*Figure 5A*); moreover, the double downregulation of SNX4 and SNX32 did not show any significant difference in the percentage of transferrin colocalized with EEA1 marked early endosomes (*Figure 5B*). Further, we went ahead to examine whether SNX32 can compensate for the loss of SNX4. We observed no significant change in the transferrin colocalization with the EEA1 compartment in SNX4KD cells overexpressing GFP-SNX32 (*Figure 5C and D*). Based on the above observations, we propose that SNX4 and SNX32 may function in a common pathway regulating the intracellular trafficking of transferrin. The physical association of SNX4 and SNX32 may provide an explanation for the above observations.

Since Rab11 is an established regulator of transferrin trafficking, we next assessed whether the GTPase plays any role in SNX32-regulated TfR trafficking. HeLa cells treated with SCR, siRab11 (RAB11KD), and/or siSNX32 (SNX32KD) cells were serum-starved for 2 hr, incubated with 10 μg/ml of transferrin (Alexa Fluor 568 conjugated) for 30 min pulse and chased using 100 μg/ml of holotransferrin for a maximum of 30 min at 37°C. As reported earlier, the colocalization of transferrin with EEA1 was significantly higher in Rab11KD cells (*Figure 5E*). Further, the depletion of both proteins led to a significant increase in the localization of transferrin with EEA1 compared to either SNX32KD or Rab11KD (*Figure 5F*). Moreover, we observed that the overexpression of GFP-SNX32 in Rab11KD cells did not show any significant difference in the transferrin colocalization with EEA1 (*Figure 5G and H*). The above observations are suggestive of the existence of an independent pathway, in addition to the one in which SNX32 may play a role upstream of Rab11.

We also noted that the downregulation of SNX32 did not have any effect on the mRNA level of either SNX4, SNX5, SNX6, or Rab11. Similarly, the knockdown of SNX1, SNX6, SNX4, or Rab11 did not alter the mRNA expression level of SNX32 (*Figure 5I*).

## SNX32 interacts with the cargo through its PX domain

Next, we sought to understand the selectivity of association of the individual interacting partner, SNX32 and SNX4/SNX1, with the cargo TfR/CIMPR. We performed a GBP-IP using cell lysate from HeLa cells overexpressing GFP-SNX4 WT/GFP-SNX1 WT/GFP-SNX32 WT. We observed that GFP-SNX32 efficiently precipitated TfR (1.2) (*Figure 6A and B*) as well as CIMPR (0.38) (*Figure 6C*); however, precipitation of TfR with GFP-SNX4 (0.59) and precipitation of CIMPR with GFP-SNX1 (0.12) were comparably less (*Figure 6A–C*).

Earlier reports showed that SNX5 interacts with its cargo, CIMPR (*Simonetti et al., 2019*)/IncE (*Paul et al., 2017*), through its PX domain. To explore whether SNX32 also interacts with the cargo through its PX domain, we employed an affinity chromatography-based pulldown assay using a His-tagged deletion mutant of SNX32, His-SNX32ΔC(1-166aa) (*Figure 1—figure supplement 1A*). We showed that the PX domain of SNX32 contributes to the interaction with TfR/CIMPR (*Figure 6D–G*).

As previously reported, the PX domains of SNX32, SNX6, and SNX5 contain a conserved stretch of 38 amino acids (*Teasdale et al., 2001*; *Figure 6—figure supplement 1A*) that, with a helix-loop-helix fold, provides a hydrophobic groove for binding to the bipartite sorting motif present in cargo proteins that include CIMPR (*Simonetti et al., 2019*). Moreover, the conserved phenylalanine within the stretch was found crucial for maintaining the interaction between SNX5 and CIMPR (*Simonetti et al., 2019*) without influencing the dimerization with its heterodimerization partner, SNX1/SNX2 (*Simonetti et al., 2019*). We superimposed the already available crystal structure of IncE-bound SNX32 PX domain (*Chandra et al., 2019*) on CIMPR-bound SNX5 (*Simonetti et al., 2019*) structure. The analysis of the superimposed structure revealed that the conserved phenylalanine (F131 for SNX32) engages in a stacking interaction with a phenylalanine (Phe) residue of IncE (*Figure 6H*). Of note, the Inc proteins of the pathogen are known to mimic the interactions of native host proteins with the sorting machinery to establish a successful infection (*Paul et al., 2017*). Thus, the F131 of SNX32 (*Figure 6—figure supplement 1B*) is predicted to directly associate with cargo proteins containing the ESCPE-1 bipartite sorting motif. To delineate the importance of the residue, we performed a GBP-IP using lysate from HeLa cells overexpressing GFP-SNX32 WT/GFP-SNX32 F131D. As shown in *Figure 6I*, the SNX32 F131D mutant failed to precipitate CIMPR, whereas it was successful in

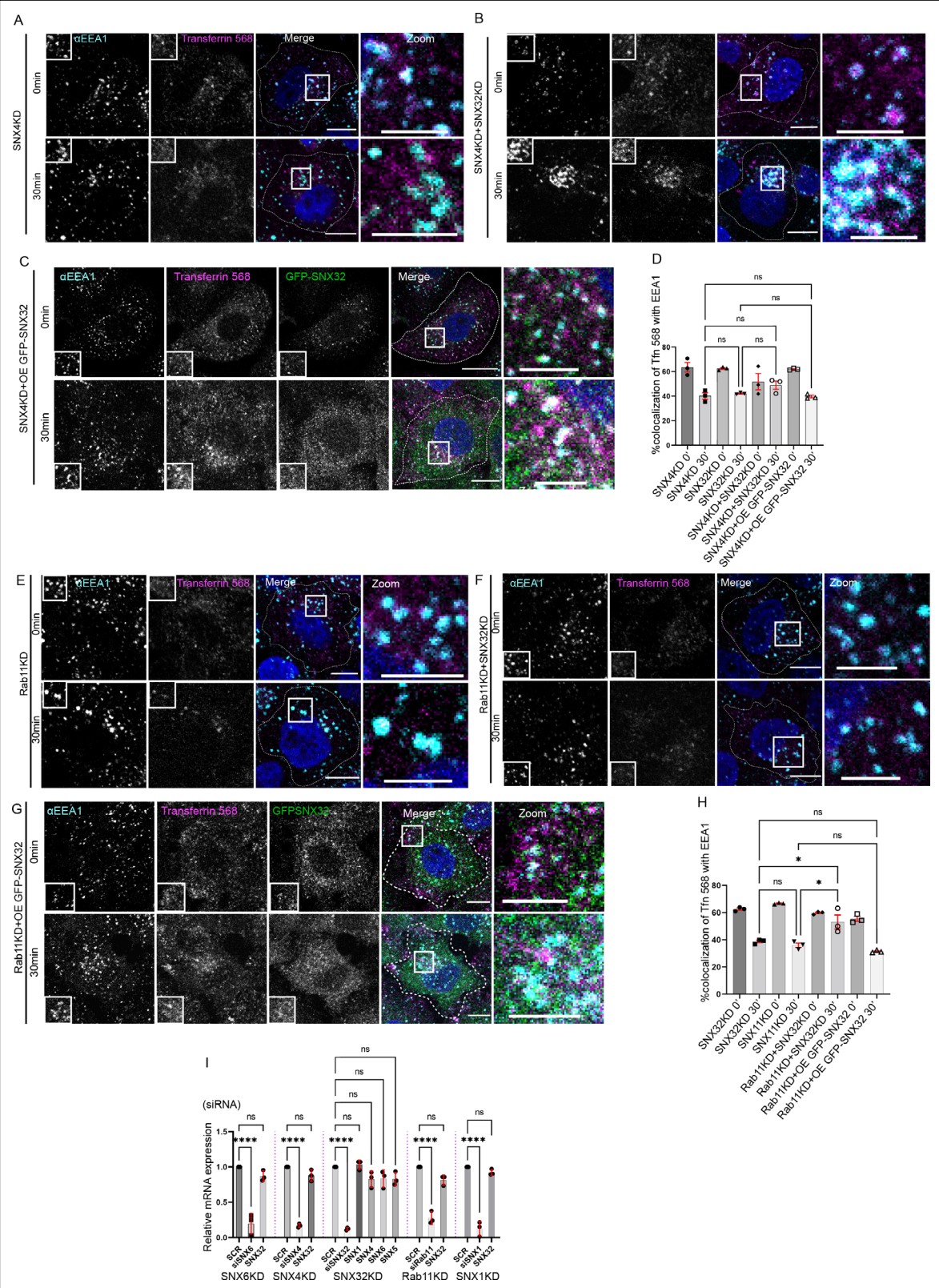

**Figure 5.** Interplay of SNX32, SNX4 and Rab11 in transferrin trafficking. (**A-C**) Followed by SMARTpoolmediated gene down regulation of (**A**) SNX4, (**B**) SNX4 and SNX32 (**C**) SNX4KD and overexpression of GFP-SNX32, the transferrin (Alexa Fluor 568 conjugated) Pulse-Chase experimentwas carried out as described in materials and method section, the cells were fixed at specifiedtimepoints, immunostained using early endosomal marker EEA1, DAPI was used to stain nucleus,Scale 10μm, inset 5μm (magnified regions are shown as insets, inset of the merge is magnified andrepresented as zoom). (**D**)

*Figure 5 continued on next page*

*Figure 5 continued*

Quantification of percentage localization of transferrin (Alexa Fluor 568conjugated) with EEA1 at corresponding time points, data represent mean ± SEM (N=3, n≥15random frames per independent experiments), P value <0.0636, Ordinary one-way ANOVA,Šídák's multiple comparisons test (ns-nonsignificant). (**E-G**) Followed by SMARTpool mediatedgene down regulation of (**E**) Rab11, (**F**) Rab11 and SNX32 (**G**) Rab11KD and over expression ofGFP-SNX32, transferrin (Alexa Fluor 568 conjugated) Pulse-Chase experiment was carried out asdescribed in materials and method section, the cells were fixed at specified timepoints,immunostained using early endosomal marker EEA1, DAPI was used to stain nucleus, Scale10μm, inset 5μm (magnified regions are shown as insets, inset of the merge is magnified andrepresented as zoom). (**H**) Quantification of percentage localization of transferrin (Alexa Fluor 568conjugated) with EEA1 at corresponding time points, data represent mean ± SEM (N=3, n≥15random frames per independent experiments), P value <0.0030, Ordinary one-way ANOVA,Šídák's multiple comparisons test (* P < 0.05, ns-nonsignificant). (**I**) Analysis of the relative geneexpression levels of SNX1,SNX5, SNX6, SNX4 and SNX32 by qRT-PCR in HeLa cells depletedfor SNX4, SNX6, SNX32, Rab11 and SNX1. Values of control were arbitrarily set as 1 againstwhich experimental data were normalized. Gapdh was used as internal control, data represent mean ± SEM (N=3) P value <0.0001, Ordinary one-way ANOVA, Šídák's multiple comparisons test(**** P < 0.0001, ns-nonsignificant).

The online version of this article includes the following source data for figure 5:

**Source data 1.** GBP co-immunoprecipitation of GFP-tagged SNX4 and SNX32 transiently transfected in HEK293T cells showing GFP-SNX32 efficiently precipitating TfR.

**Source data 2.** GBP co-immunoprecipitation of GFP-tagged SNX1 and SNX32 transiently transfected in HEK293T cells showing GFP-SNX32 efficiently precipitating CIMPR.

**Source data 3.** His affinity chromatography-based pulldown showing His-SNX32ΔC precipitating TfR from membrane-enriched HeLa cell lysate fraction.

**Source data 4.** His affinity chromatography-based pulldown showing His-SNX32ΔC precipitating CIMPR from membrane-enriched HeLa cell lysate fraction.

**Source data 5.** Co-immunoprecipitation of GFP-tagged SNX32 wild-type (WT) and GFP trap of GFP-tagged SNX32 WT/SNX32 F131D, showing both efficiently precipitating ESCPE-1 subunit SNX1, whereas SNX32 F131D failed to precipitate CIMPR, each transiently transfected in HEK293T cells.

**Source data 6.** GBP co-immunoprecipitation of GFP-tagged SNX-proteins transiently transfected in HeLa cells showing GFP-SNX32 but not GFP-SNX32 F131D efficiently pulling down TfR.

precipitating endogenous SNX1, thereby validating the role of the Phe in binding with CIMPR. We further investigated whether the interaction between SNX32 and TfR also follows a similar mechanism. The GBP-IP of GFP-SNX32 WT/GFP-SNX32 F131D showed that the F131D mutant of SNX32 failed to precipitate TfR (*Figure 6J and K*), affirming the role of F131 in TfR binding.

Though our investigations on SNX32 revealed its involvement in the intracellular trafficking of two well-established cargoes in HeLa cells, the relatively higher expression of SNX32 in brain tissue made us inquisitive about its significance in neuroglial cells.

## SNX32 plays a crucial role in regulating neurite outgrowth

To explore the functional implications of SNX32 in brain tissue, we focused on neuronal differentiation. Neurite outgrowth formation, projection, and extension of neural processes are fundamental in normal neuronal development. We utilized neurite outgrowth assay (*Ma et al., 2015*), which serves as an in vitro model to investigate the potential effects of any molecule of interest in neuronal differentiation. Neuro2a (a mouse neuroblastoma cell line) cells treated with siRNAs targeting SNX32 (SNX32KD) and SNX6 (SNX6KD) (*Figure 7—figure supplement 1A*) were differentiated to form neurites by replacing the standard growth media (MEM with 10% FBS) with the differentiation media (MEM with 1% FBS containing 10 μmol/l retinoic acids) and incubated for 48 hr. We observed that in SNX32KD cells, the number of cells with elongated neurites was less compared to SCR or SNX6KD conditions (*Figure 7A and B*). Even though most of the cells in the SNX32KD condition showed positive sprouting, the length of these sprouts was limited (less than the diameter of the cell body). To better understand and negate any artifacts introduced due to cell fixation, the procedure of neurite induction was replicated, and images were captured in real time for a maximum of 48 hr. The DIC time-lapse live-cell video imaging (*Video 5*) showed that the response to retinoic acid (RA) (supplemented in 1% FBS containing MEM media) was prompt and rapid. Sprouting and normal growth cone movement were visible in ≥80% of the cells within the initial few hours in all the conditions under investigation (*Figure 7C–E*). As time progressed, in SCR and SNX6KD cells, long neurites became more prevalent and intermingled with adjacent cell neurites, forming intricate networks (*Figure 7F and G*). Whereas in the case of the SNX32KD condition, the sprouts failed to elongate and were inefficacious in establishing the neural network (*Figure 7H*). However, it is interesting to note that in either case of SNX32KD/SNX6KD conditions, cell proliferation was much slower than in the SCR condition (*Video 5*).

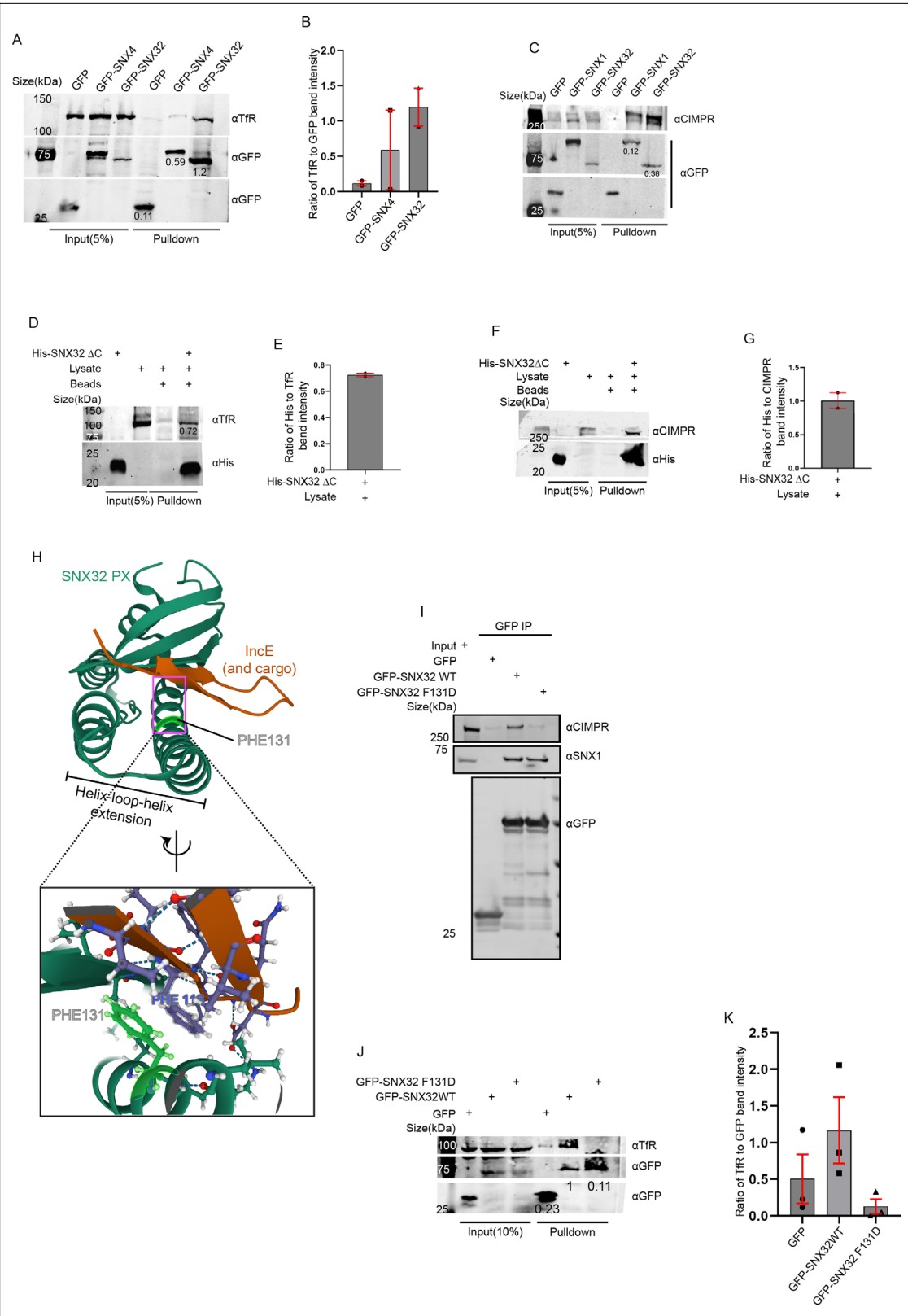

**Figure 6.** PX domain of SNX32 participates in the interaction with CIMPR and TfR, and F131 of SNX32 is critical for interaction with cargo. (**A**) GBP co-immunoprecipitation of GFP-tagged SNX4 and SNX32 transiently transfected in HEK293T cells showing GFP-SNX32 efficiently precipitating TfR. GBP immunoprecipitation was carried out as described in the 'Materials and methods' section and immunoblotted using GFP and TfR antibody (representative immunoblot out of three biological replicates, values represent the ratio of TfR to GFP band intensity). (**B**) Plot representing

*Figure 6 continued on next page*

*Figure 6 continued*

quantifications of the three independent experiments of GBP co-immunoprecipitation of GFP/HA-tagged SNX-proteins represented in (**A**). (**C**) GBP co-immunoprecipitation of GFP-tagged SNX1 and SNX32 transiently transfected in HEK293T cells showing GFP-SNX32 efficiently precipitating CIMPR. GBP immunoprecipitation was carried out as described in the 'Materials and methods' section and immunoblotted using GFP and TfR antibody (representative immunoblot out of three biological replicates, values represent the ratio of CIMPR to GFP band intensity). (**D**) His affinity chromatography-based pulldown showing His-SNX32ΔC precipitating TfR from membrane-enriched HeLa cell lysate fraction. His pulldown was carried out as described in the 'Materials and methods' section and immunoblotted using His and TfR antibody (representative immunoblot out of three biological replicates, values represent the ratio of His to TfR band intensity). (**E**) Plot representing quantifications of the two independent experiments of GBP co-immunoprecipitation of GFP/HA-tagged SNX-proteins represented in (**D**). (**F**) His affinity chromatography-based pulldown showing His-SNX32ΔC precipitating CIMPR from membrane-enriched HeLa cell lysate fraction. His pulldown was carried out as described in the 'Materials and methods' section and immunoblotted using His and TfR antibody (representative immunoblot out of three biological replicates). (**G**) Plot representing quantifications of the two independent experiments of GBP co-immunoprecipitation of GFP/HA-tagged SNX-proteins represented in (**F**). (**H**) Cargo/IncE-binding site of SNX32 PX domain as observed in crystal structure (PDB ID: 6E8R) reported by *Chandra et al., 2019*, inset showing the stacking interaction between F131 of SNX32 and F116 of IncE. (**I**) Co-immunoprecipitation of GFP-tagged SNX32 wild-type (WT) and GFP trap of GFP-tagged SNX32 WT/ SNX32 F131D, showing both efficiently precipitating ESCPE-1 subunit SNX1, whereas SNX32 F131D failed to precipitate CIMPR, each transiently transfected in HEK293T cells. The elute was resolved in SDS-PAGE and immunoblotted using GFP, SNX1, and CIMPR antibody. (**J**) GBP co-immunoprecipitation of GFP-tagged SNX-proteins transiently transfected in HeLa cells showing GFP-SNX32 but not GFP-SNX32 F131D efficiently pulling down TfR. GBP immunoprecipitation was carried out as described in the 'Materials and methods' section and immunoblotted using GFP and TfR antibody (representative immunoblot out of three biological replicates, values represent the ratio of TfR to GFP band intensity). (**K**) Plot representing quantifications of the three independent experiments of GBP co-immunoprecipitation of GFP /HA-tagged SNX-proteins represented in (**J**).

The online version of this article includes the following source data and figure supplement(s) for figure 6:

**Source data 1.** Co-immunoprecipitation of mCherry-tagged SNX-proteins transiently transfected in U87MG cells showing mCherry-SNX32 but not mCherry-SNX6 efficiently pulling down Basigin (BSG).

**Source data 2.** Histidine (His) pulldown showing His SNX32ΔC efficiently pulling down BSG from membrane-enriched HeLa lysate.

**Source data 3.** GFP-tagged SNX-proteins transiently transfected in HeLa cells showing GFP-SNX32 WT but not GFP-SNX32 F131D efficiently pulling down Basigin (BSG).

**Figure supplement 1.** Mapping the interactome of SNX32, utilizing the conserved F131 residue.

## Differential proteomics implied BSG as an interacting partner of SNX32

During neuronal differentiation and polarity establishment, the directional intracellular transport maintains the supply and distribution of molecules to the growing processes (*Matsuzaki et al., 2011*). In order to investigate the role of SNX32 in the directional flow, we set out to identify the cargo repertoire (*Simonetti et al., 2017*) of the SNX. We applied a quantitative SILAC-based proteomics approach to define the interactome in neuronal cells. We first lentivirally transduced SHSY5Y, a human neuroblastoma cell line to stably express GFP-tagged SNX32. In agreement with the previous report in RPE1 cells (*Simonetti et al., 2017*), in SHSY5Y cells, GFP-SNX32 largely colocalized with EEA1 harbouring endosomes (*Figure 7I*). The endosomal localization of GFP-SNX32 with EEA1 and in addition, Rab5 was also observed in U87MG cells (*Figure 7J*).

As previously described, the F131D mutant of SNX32 fails to interact with cargoes such as CIMPR (*Figure 6I*) and TfR (*Figure 6J and K*). So, we further established an SHSY5Y cell line stably expressing GFP-SNX32 F131D. We observed that GFP-SNX32 F131D also retained the ability to associate with endosomes (*Figure 7—figure supplement 1B*). We then performed differential SILAC-based proteomics by comparing the interactomes of wild-type SNX32 and SNX32 F131D (*Figure 7—figure supplement 1C*). The comparative proteomics highlighted several cargoes that bind to SNX32 through a mechanism that requires the critical F131 residues within the hydrophobic groove of the SNX32 PX domain. Based on the abundance ratio: (SNX32 F131D)/(SNX32 WT), we found several transmembrane proteins interacting with SNX32, including the well-established cargo CIMPR. Among these, BSG was one of the strongest quantified interactors in addition to CIMPR. BSG showed enrichment in SNX32 interactome but a drastic reduction in F131D interactome (*Figure 7K*). BSG is a transmembrane receptor belonging to the superfamily of immunoglobulins. BSG, also known as EMMPRIN, CD147, or human leukocyte activation-associated M6 antigen, is involved in a myriad of cellular processes, including tissue remodeling, visual sensory system development in mice, regulation of neuronal development, and the cell surface concentration of monocarboxylate transporters (MCTs) (*Muramatsu, 2016*). To validate the interaction of SNX32 with BSG, we first utilized an mCherry nanobody-based immunoprecipitation (mCBP-IP) (*Fridy et al., 2014*) assay. GST-tagged mCBP

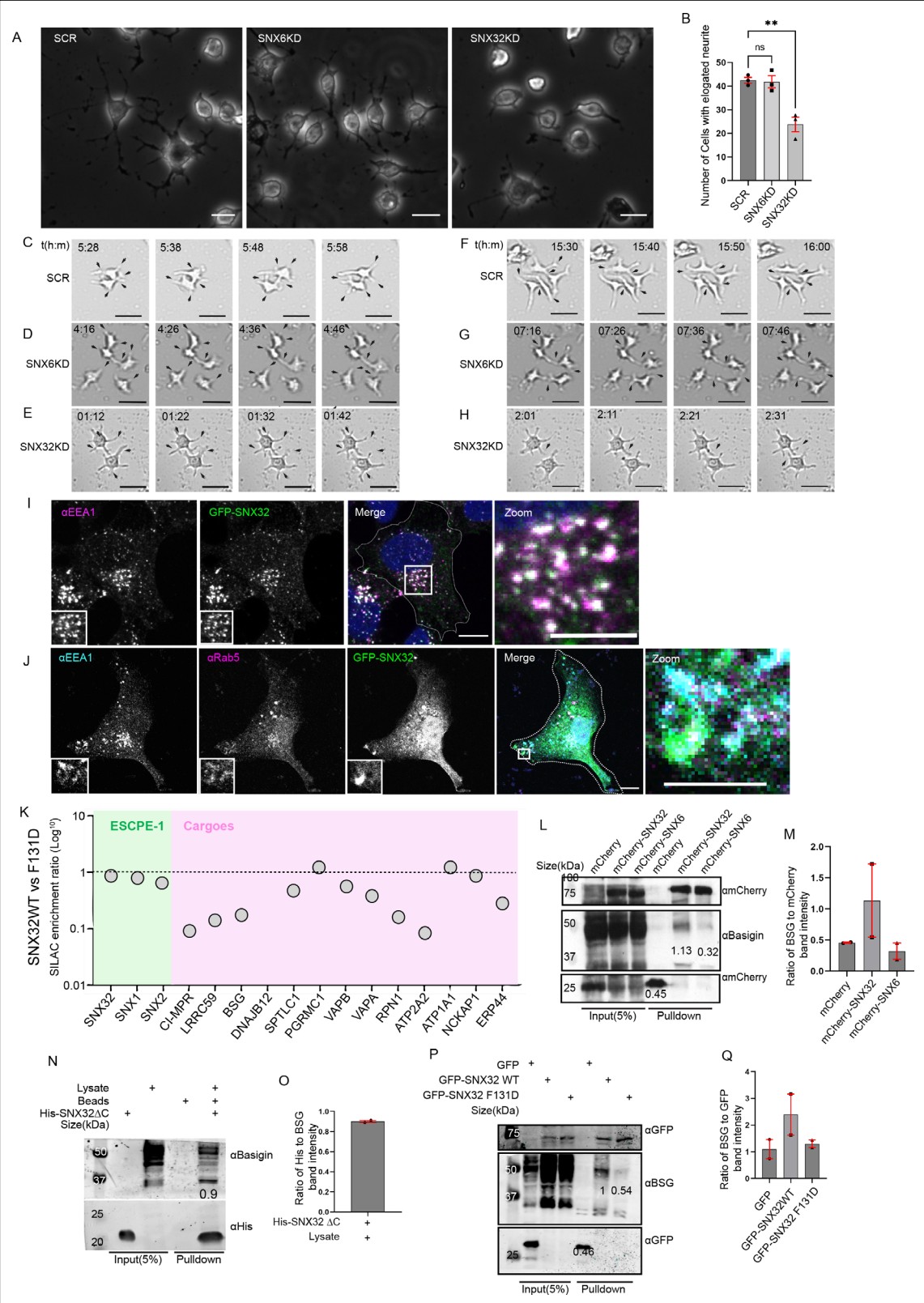

**Figure 7.** SNX32 plays a crucial role in neurite differentiation, and it interacts with immunoglobulin superfamily member BSG via its PX domain. (**A**) Phase-contrast image of Neuro2a cells transfected with scrambled (SCR)/SNX32/SNX6 siRNA SMART pool followed by neurite induction as described in the 'Materials and methods' section, fixed, and imaged using Zeiss Axio vert. A1 microscope. Scale bar 50 µm. (**B**) Quantification of number of cells with elongated neurites, data represent mean ± SEM (N = 3, n ≥ 100 cells per independent experiments, values are means ± SEM), p-value

*Figure 7 continued on next page*

*Figure 7 continued*

0.0025 (**p<0.01), ordinary one-way ANOVA Dunnett's multiple-comparisons test. (**C–E**) Snapshots of SCR/SNX6/SNX32 SMART pool siRNA-transfected Neuro2a cells induced with RA for neurite induction showing neurite sprouting. Scale bar 50 µm (black arrows pointing to sprouting). (**F–H**) Snapshots of Neuro2a cells transfected with SCR/SNX6/SNX32 SMART pool siRNA-transfected Neuro2a cells induced with RA for neurite induction showing neurite extension, black arrows pointing to network formation/retraction of neurites. Scale bar 50 µm. (**I**) SHSY5Y cells showing colocalization of GFP-tagged SNX32 with EEA1 harboring endosomes. Scale bar 10 µm, inset 5 µm (magnified regions are shown as insets, inset of the merge is magnified and represented as zoom). (**J**) U87MG cells showing the colocalization of GFP-tagged SNX32 with Rab5 and EEA1 harboring endosomes. Scale bar 10 µm, inset 5 µm (magnified regions are shown as insets, inset of the merge is magnified and represented as zoom). (**K**) SILAC enrichment ratio (Log10) of SNX32 WT vs SNX32 F131D. SILAC was carried as described in the 'Materials and methods' section, the proteins consistently appearing in at least two experimental replicates were considered for the final analysis. (**L**) Co-immunoprecipitation of mCherry-tagged SNX-proteins transiently transfected in U87MG cells showing mCherry SNX32 but not mCherry SNX6 efficiently pulling down Basigin (BSG). mCherry nanobody-mediated pulldown was carried out as described in the 'Materials and methods' section and immunoblotted using mCherry and BSG antibody (representative immunoblot out of three biological replicates, values represent the ratio of BSG to mCherry band intensity). (**M**) Plot representing quantifications of the three independent experiments of GBP co-immunoprecipitation of GFP/HA-tagged SNX-proteins represented in (**L**). (**N**) Histidine (His) pulldown showing His SNX32ΔC efficiently pulling down BSG from membrane-enriched HeLa lysate (representative immunoblot out of three biological replicates, values represent the ratio of BSG to His band intensity). (**O**) Plot representing quantifications of the three independent experiments of GBP co-immunoprecipitation of GFP/HA-tagged SNX-proteins represented in (**N**). (**P**) GFP-tagged SNX-proteins transiently transfected in HeLa cells showing GFP SNX32 WT but not GFP SNX32 F131D efficiently pulling down Basigin (BSG). GFP nanobody-mediated pulldown was carried out as described in the 'Materials and methods' section and immunoblotted using GFP and BSG antibody (representative immunoblot out of three biological replicates, values represent the ratio of BSG to GFP band intensity). (**Q**) Plot representing quantifications of the three independent experiments of GBP co-immunoprecipitation of GFP/HA-tagged SNX-proteins represented in (**P**).

The online version of this article includes the following figure supplement(s) for figure 7:

**Figure supplement 1.** F131D mutation does not alter the endosomal localization of GFP-SNX32.

immobilized on glutathione sepharose beads was incubated with U87MG cell extracts overexpressing mCherry/mCherry-SNX6/mCherry-SNX32. Immunoblotting using an anti-BSG antibody showed that endogenous BSG was efficiently precipitated by mCherry-SNX32 but not mCherry-SNX6 (*Figure 7L and M*). As previously demonstrated, the PX domain of SNX32 precipitates CIMPR and TfR. To identify whether a similar mechanism is relevant in the case of interaction with BSG, we performed an affinity chromatography-based pulldown assay, which showed that the PX domain of SNX32 contributes to the interaction with BSG (*Figure 7N and O*). We also carried out GBP-IP of full-length GFP- SNX32 WT and GFP-SNX32 F131D. While the endogenous BSG protein could be co-immunoprecipitated with the GFP-SNX32 wild-type, the interaction was lost for the mutant (*Figure 7P and Q*). Together, these data establish BSG as a selective interacting partner for canonical binding to the PX domain of the ESCPE-1 component SNX32.

## SNX32 phenocopies BSG in neurite outgrowth assay

Previously, it was reported that BSG plays a crucial role in complex (*Shrestha et al., 2021*) and space-filling dendrite growth (*Alizzi et al., 2020*) in *Drosophila*. As we observed a similar phenotype in SNX32-suppressed conditions, we next set forth to see whether the case is similar under BSG downregulation in neurite induction of Neuro2a cells. shRNA clones were used to downregulate SNX32 (SNX32KD) or BSG (BSGKD) (*Figure 8—figure supplement 1A and B*) and the cells were differentiated to form neurites by replacing the standard growth media (MEM with 10% FBS) with the differentiation media (MEM with 1% FBS containing 10 µmol/l retinoic acid). We observed that in BSG-deficient cells the number of cells with elongated neurites was less, which is in accordance with the previous reports (*Lagenaur et al., 1992*). Further, we noted that the number of cells with elongated neurites in the case of SNX32KD was comparable to that of BSGKD (*Figure 8A and B*, *Video 6*), suggesting that the gene silencing

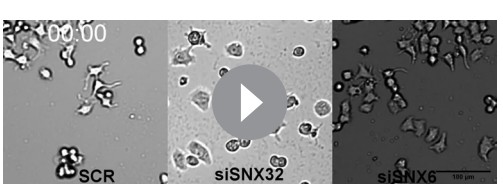

**Video 5.** SNX32 depletion hinders neurite outgrowth and network formation. Neuro2a cells were transfected with scrambled/SNX32/SNX6 siRNA SMART pools. Following 24 hr of transfection, the medium was replaced with MEM containing 1% fetal bovine serum supplemented with 10 µmol/l retinoic acid (RA) to induce neurite outgrowth. Frames were captured every 1 min for 48 hr using a ×4 objective and a CMOS camera utilizing incubator-compatible JuLIBr Live cell analyzer. Playback rate 24 frames/s.

https://elifesciences.org/articles/84396/figures#video5

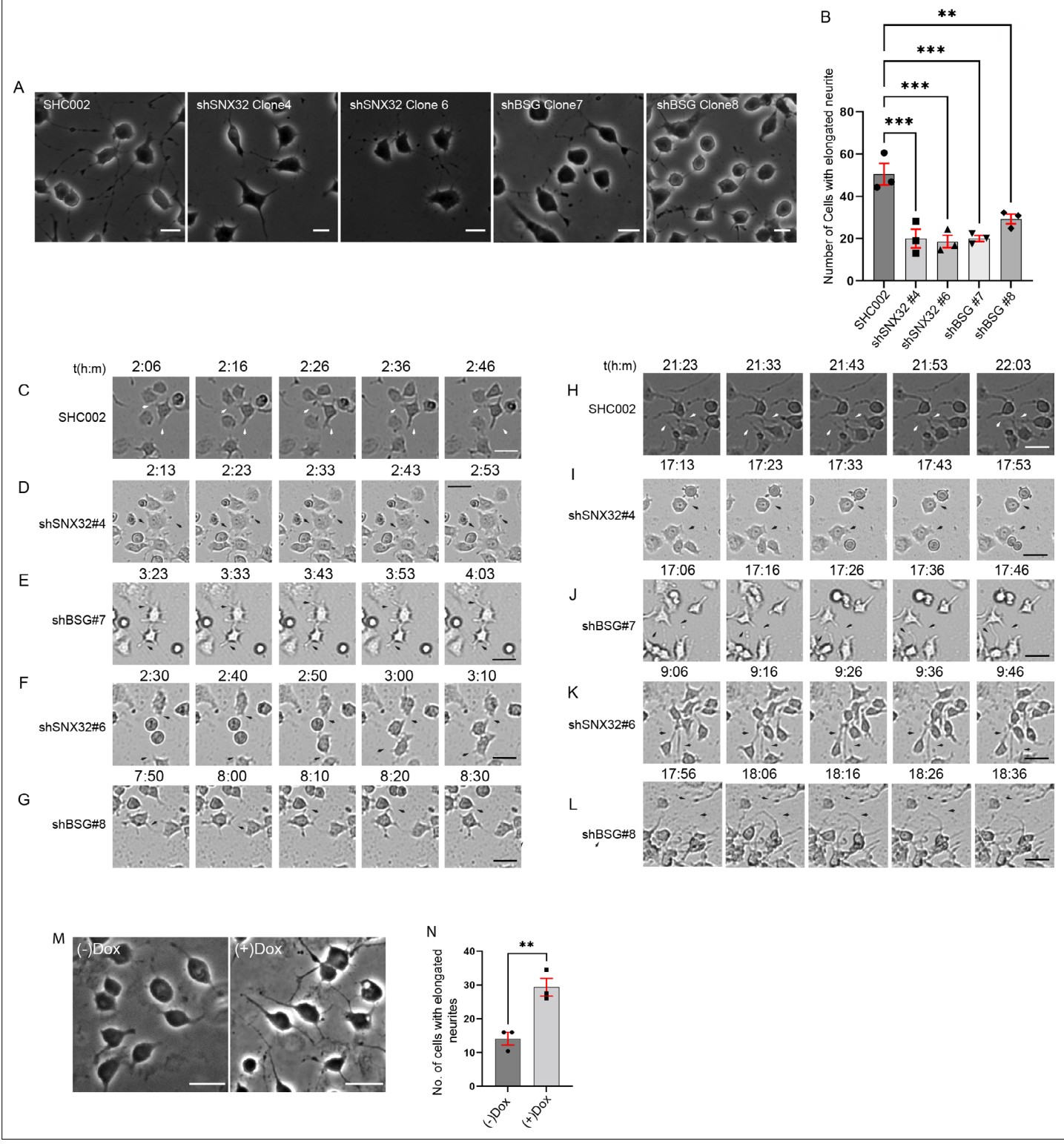

**Figure 8.** SNX32 and BSG play a crucial role in neurite network formation. (**A**) Phase-contrast images of Neuro2a cells, which were transfected with scrambled (SHC002)/SNX32 (shSNX32#4, shSNX32#6)/BSG (shBSG#7, shBSG#8) shRNA clones followed by neurite induction as described in the 'Materials and methods' section, fixed, and imaged using Zeiss Axio vert. A1 microscope. Scale bar 50 μm. (**B**) Quantification of number of cells with elongated neurites, data represent mean ± SEM (N = 3, n ≥ 100 cells per independent experiments, values are means ± SEM), p-value 0.0003 (**p<0.01, ***p<0.001), ordinary one-way ANOVA Dunnett's multiple-comparisons test. (**C–G**) Snapshots of Neuro2a cells transfected with scrambled (SHC002)/SNX32 (shSNX32#4, shSNX32#6)/BSG (shBSG#7, shBSG# 8) shRNA clones followed by neurite induction with RA for neurite induction

*Figure 8 continued on next page*

showing neurite sprouting. Scale bar 50 μm (black arrows pointing to sprouting). (**H–L**) Snapshots of Neuro2a cells transfected with scrambled (SHC002)/ SNX32 (shSNX32#4, shSNX32#6)/BSG (shBSG#7, shBSG# 8) shRNA clones followed by neurite induction with RA for neurite induction showing neurite extension, black arrows pointing to network formation/retraction of neurites. Scale bar 50 μm. (**M**) Phase-contrast image of Neuro2a cells stably expressing pLVX SNX32 transfected with shSNX32#4 followed by neurite induction with or without doxycyclin as described in the 'Materials and methods' section, fixed, and imaged using Zeiss Axio vert. A1 microscope. Scale bar 50 μm. (**N**) Quantification of number of cells with elongated neurites, data represent mean ± SEM (N = 3, n ≥ 100 cells per independent experiments), p-value 0.0091 (**p<0.01), unpaired t test.

The online version of this article includes the following figure supplement(s) for figure 8:

**Figure supplement 1.** Relative mRNA expression quantification using RT-PCR.

of SNX32 phenocopies the depletion of BSG (*Lagenaur et al., 1992*). Although most cells showed sprouting in either SNX32 or BSG-suppressed conditions, the length of these sprouts was limited (less than the diameter of the cell body). As evident from the DIC time-lapse live-cell video imaging, sprouting was visible in ≥80% of the cells within the initial few hours in all conditions (*Figure 8C–G*). In control cells, long neurites were prevalent, and they intermingled with adjacent cell's neurites to form intricate networks (*Figure 8H*). In contrast, upon SNX32 or BSG suppression mediated by shRNA clones shSNX32#4 or shBSG#7, sprouts failed to elongate, and they were inefficacious in establishing the neural network (*Figure 8I and J*), while in cells transfected with shSNX32#6 or shBSG#8 the neural tubes elongated but without branching or the formation of any network (*Figure 8K and L*). Interestingly, in neither case, the growth cone movement was affected (*Video 6*), which is consistent with the perturbation of neurite extension observed upon treatment with an antibody against BSG (*Lagenaur et al., 1992*). Taken together, these observations suggest that SNX32 is functionally linked with BSG in neuronal development.

To validate our observations and further ensure the gene-specific effect, we performed a rescue experiment utilizing shRNA-resistant SNX32 construct (pLVX shSNX32#4r). Neuro2a cells stably expressing doxycycline-inducible pLVX shSNX32#4r were transfected with shSNX32#4 (SNX32KD) to acutely deplete endogenous SNX32. The SNX32KD cells were differentiated to form neurites by replacing the standard growth media (MEM with 10% FBS) with the differentiation media (MEM with 1% FBS containing 10 μmol/l retinoic acid). The cells were allowed to grow in differentiation media with or without doxycycline for the next 48 hr. Further, the cells were fixed, phase-contrast images were acquired, and cells with elongated neurites were counted. We observed that

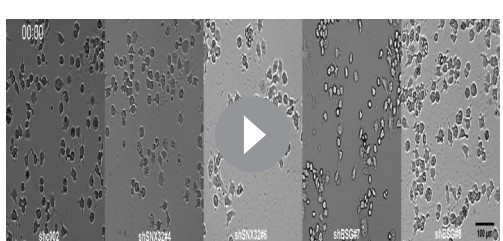

**Video 6.** SNX32 depletion phenocopies the neurite outgrowth defect observed in BSG downregulated condition. Neuro2a cells were transfected with SHC002/ shSNX32 (#4,#6)/shBSG clones (#7,#8). Following 6 hr (shRNA) of transfection, the medium was replaced with MEM containing 1% fetal bovine serum supplemented with 10 μmol/l retinoic acid (RA) to induce neurite outgrowth. Frames were captured every 1 min for a maximum of 48 hr using a ×4 objective and a CMOS camera utilizing incubator-compatible JuLIBr Live cell analyzer. Playback rate 24 frames/s.

https://elifesciences.org/articles/84396/figures#video6

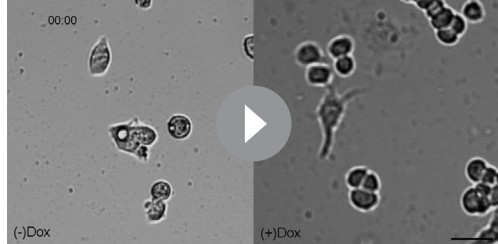

**Video 7.** shRNA-resistant SNX32, shSNX32#4r overexpression could rescue the neurite outgrowth defect observed in BSG downregulated condition. Neuro2a cells stably transfected with pLVXshSNX32#4r were transfected with shSNX32#4. Following 6 hr (shRNA) of transfection, the medium was replaced with MEM containing 1% fetal bovine serum supplemented with 10 μmol/l retinoic acid (RA) to induce neurite outgrowth. Frames were captured every 15 min for a maximum of 48 hr using a ×4 objective and a CMOS camera utilizing incubator-compatible JuLIBr Live cell analyzer. Playback rate 3 frames/s.

https://elifesciences.org/articles/84396/figures#video7

the doxycycline-supplemented cells showed an increased number of cells with elongated neurites compared to no doxycycline-supplemented cells (*Figure 8M and N*, *Video 7*).

## SNX32 is essential for the surface trafficking of BSG

Next, we sought to assess whether the surface population of BSG is altered in SNX32 downregulated conditions using TIRF microscopy. We utilized a Neuro2a cell line stably expressing an inducible vector pLVX TRE3G containing pHluorin (*Miesenböck et al., 1998*) tagged BSG. We observed that compared to scrambled (SCR) siRNA (*Figure 9A*), the downregulation of SNX32 (SNX32KD) in Neuro2a cells substantially reduced the surface population of pHluorin-BSG (*Figure 9B*, *Video 8*) Moreover, we validated our observation using shRNA-mediated SNX32 downregulation (*Figure 9—figure supplement 1A–D*, *Video 9*). Interestingly SNX6KD did not show any effect (*Figure 9C and D*), consistent with the absence of interaction between SNX6 and BSG. Furthermore, immunoblot analysis showed that in SNX32-silenced conditions the protein levels of BSG were subtly reduced compared to Scrambled (SCR/SHC002) or SNX6KD conditions (*Figure 9—figure supplement 1E and F*).

## SNX32 downregulation disrupts the MCT-mediated lactate shuttling

Neuroglial coordination is necessary for the establishment of neuronal networks. The constant shuttling of metabolic fuels such as ketone bodies, pyruvate, and lactate (*Bergersen, 2007*) plays a major role in contributing to this coordination. Monocarboxylic acid transporters (MCTs) play a critical role in neuroglial coordination (*Jha and Morrison, 2020*) by facilitating the transport of monocarboxylates (*Gallagher et al., 2007*). It has been reported that BSG acts as a cochaperone for MCTs and facilitates their surface localization (*Kirk et al., 2000*). Accordingly, BSG knockdown causes the accumulation of MCTs in the endolysosomal compartment, leading to reduced lactate concentration in the culture supernatant (*Walters et al., 2013*). Lactate is the most common substrate for MCT transport in the brain, where it is transported from its site of synthesis to the site of consumption between the glia and neurons (*Bergersen, 2015*). Accordingly, the reduced lactate shuttle between neuroglial cells interferes with the neurite outgrowth (*Chen et al., 2018*).

Therefore, we asked whether SNX32 regulates MCT activity via its role in BSG trafficking. When expressed from a transiently transfected inducible vector in U87MG cells, GFP-SNX32 showed considerable colocalization with BSG (*Figure 9—figure supplement 1G*). As previously reported, a significant population of BSG colocalized with ARF6 at the cell surface (*Eyster et al., 2009*; *Figure 9E and F*). Similarly, in Neuro2a cells stably expressing cMyc-tagged BSG, there was considerable colocalization with GFP-SNX32 on intracellular punctae (*Figure 9G*), which was further confirmed by live-cell video microscopy (*Video 10*).

We next sought to quantify the extracellular concentration of lactate in cultured U87MG cells under SNX32-depleted condition. We performed a lactate quantification assay under SNX32KD or BSGKD conditions. The knockdown efficiency obtained by employing siRNA or shRNA in U87MG cells was measured at the mRNA level using qRT-PCR (*Figure 9—figure supplement 1H–J*). The culture supernatant of SNX32KD/BSG KD cells was collected and the amount of lactate was quantified following the manufacturer's protocol. As reported earlier, BSG depletion led to reduced lactate concentration in the culture supernatant. We observed a similar reduction of lactate concentration in SNX32 downregulated condition (*Figure 9H*, *Figure 9—figure supplement 1K*). In contrast, the SNX6KD, the paralogue of SNX32, did not show any significant effect (*Figure 9H*). Our results suggest that in addition to its functional link with BSG, SNX32 is important for maintaining the activity of MCTs.

SNX32, therefore, regulates the cell surface trafficking of BSG, consistent with a working hypothesis that the role of SNX32 in neurite development is in part attributed to its ability to regulate the trafficking of BSG and, thereby, MCT.

## Discussion

The evolutionarily conserved SNX-BAR family of proteins are implicated in cargo identification, sorting, and membrane tubule biogenesis (*Teasdale and Collins, 2012*; *Gallon and Cullen, 2015*). The co-dependence of SNXs owing to the ability of SNX-BARs to undergo homo/hetero-dimerization is a critical feature during various stages of cargo sorting (*Simonetti et al., 2017*; *Niu et al., 2013*; *Seaman, 2004*; *Bonifacino and Hurley, 2008*; *Hong et al., 2009*). In HeLa cells, the results from the

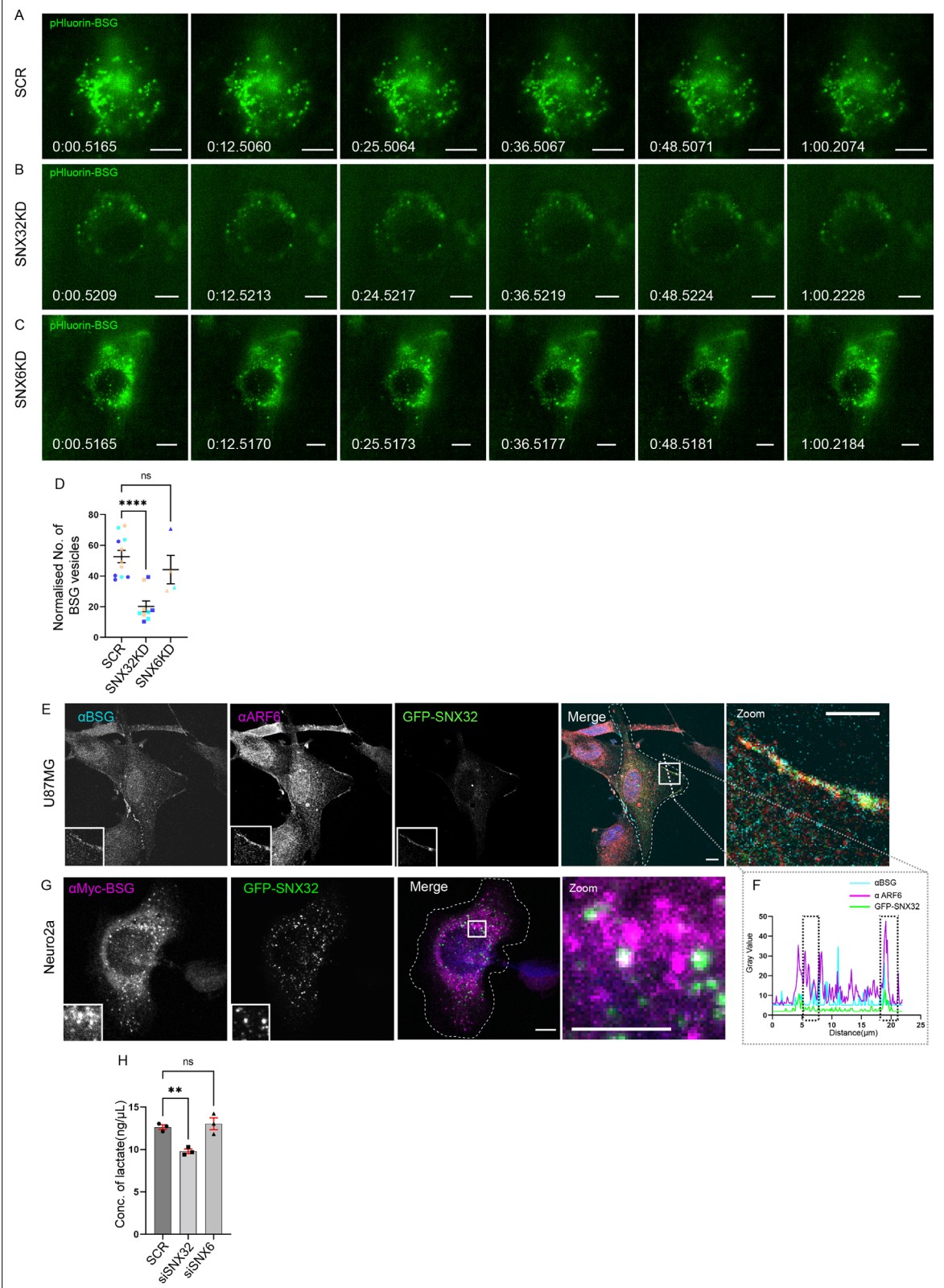

**Figure 9.** SNX32 but not SNX6 plays a significant role in surface localization of BSG. (**A–C**) Snapshots from live TIRF microscopic imaging of Neuro2a cells stably expressing pHluorin BSG transfected with (**A**) SCR, (**B**) SNX32, or (**C**) SNX6 siRNA SMART pool followed by doxycycline treatment for pHluorin BSG induction. Scale bar 10 μm. (**D**) Quantification of surface population of normalized number of BSG vesicles (N = 3, n ≥ 6 cells per independent experiments), represented as a SuperPlot20, prepared by superimposing summary statistics from repeated experiments on a graph of

*Figure 9 continued on next page*

*Figure 9 continued*

the individual measurements per cell. The samples per experiment are represented by color-coding the dots and the mean from each experiment is plotted as an error bar on top of the many smaller dots that denote individual measurements. The error bar is applied on the mean from each individual experiment. p-value 0.0003 (****p<0.0001, ns, nonsignificant), ordinary one-way ANOVA Dunnett's multiple-comparisons test. (**E**) U87MG cells showing the colocalization of GFP-SNX32 with endogenous ARF6 and BSG on membrane. Scale bar 10 µm, inset 5 µm (magnified regions are shown as insets, inset of the merge is magnified and represented as zoom). (**F**) Representative line intensity plot of U87MG cell showing intensity overlap of GFP-SNX32, ARF6, and BSG. (**G**) Neuro2a cells showing colocalization of GFP SNX 32 with cMyc-BSG on vesicles. Scale bar 10 µm, inset 5 µm (magnified regions are shown as insets, inset of the merge is magnified and represented as zoom). (**H**) Quantification of concentration of lactate in the culture supernatant of U87MG cells transfected with SCR/SNX32/SNX6 SMART pool siRNA, N = 3, values are means ± SEM, p-value 0.0049 (**p<0.01, ns, nonsignificant), ordinary one-way ANOVA Dunnett's multiple-comparisons test.

The online version of this article includes the following source data and figure supplement(s) for figure 9:

**Figure supplement 1.** SNX32 regulates the surface population of BSG.

**Figure supplement 1—source data 1.** U87MG cells were transfected with scramble/SNX32/SNX6 siRNA or shc002, shSNX32#4, shSNX32#6 shRNA followed by cycloheximide treatment of 10 µg/ml for 6 hr.

colocalization studies on SNX32 in the absence or presence of SNX1/SNX4 (*Figure 3A*, *Figure 4A*, *Figure 2—figure supplement 1D and E*) or the PI3K inhibitor Wortmannin (*Figure 2—figure supplement 1K–M*) indicate that the BAR domain-mediated interaction of the SNX with SNX1/SNX4 may contribute to its recruitment to EEA1, harboring early endosomes. Similar interdependence was also observed for SNX6 and SNX27, where their heterodimerization with SNX1/2 or SNX1, respectively, was shown to be important for membrane recruitment (*Hong et al., 2009*; *Yong et al., 2021*). Though the contribution of heteromeric interactions in membrane localization of SNX32 is in place, the role of PIns affinity of the PX domain cannot be disregarded. The results from our cellular localization study on SNX32-PX in the presence or absence of Wortmannin (*Figure 2—figure supplement 1L*) indicated that SNX32's PX domain might also contribute to its localization on endosomes. Likewise, its affinity toward PI4P, as evident from its localization in PAO-treated cells (*Figure 2K*), corroborates well with SNX32's association with Golgi/recycling compartments. Although this was further supported by the results from the PIP strip-based studies (*Figure 2—figure supplement 2A*) using His-SNX32ΔC, the PLiMAP investigations did not show any detectable binding of His-SNX32ΔC to PI(3)P or PI(4)P (*Figure 2—figure supplement 2B*). However, the apparent contradiction of the above observations could be explained by SNX32's low PIns binding affinities. Based on these observations, we hypothesize that as reported earlier for SNX1 (*Stenmark et al., 2002*)/SNX4 (*Seaman et al., 2009*), both the PX domain as well as the BAR domain contribute to the membrane recruitment of SNX32 (*Carlton et al., 2004*; *Traer et al., 2007*).

Moreover, the cooperative binding of the heterodimeric partners could also be crucial in the membrane remodeling as well as cargo recognition stages of cargo sorting. Earlier reports have shown that SNX32, similar to its closest homologs SNX5 and SNX6, was unable to induce in vitro membrane tubulation *van Weering et al., 2012*; our results indicate that it may not be due to the SNX's insufficiency in membrane association. Based on our results, we hypothesize that SNX32 engages in a heteromeric interaction with a partner, efficient in inducing membrane tubules such as SNX4, while

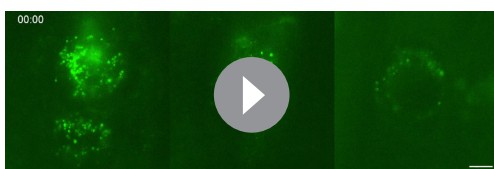

**Video 8.** Surface population of pHluorin BSG is reduced in SNX32-deficit condition. Neuro2a cells stably expressing TET-inducible pHluorin BSG were transfected with scrambled/SNX32/SNX6 siRNA SMART pools for a maximum of 72 hr. 13 hr prior to imaging the cells were induced with doxycycline for pHluorin BSG induction. Frames were collected every 3.3 s for 1 min. Playback rate is 3 frames/s.
https://elifesciences.org/articles/84396/figures#video8

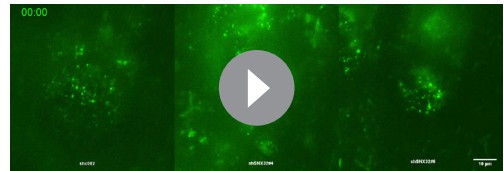

**Video 9.** Surface population of pHluorin BSG is reduced in SNX32-deficit condition. Neuro2a cells stably expressing TET-inducible pHluorin BSG were transfected with SHC002/SNX32(#4, #6) shRNA clones for a maximum of 42 hr. 13 hr prior to imaging, the cells were induced with doxycycline for pHluorin BSG induction. Frames were collected every 3.3 s for 1 min. Playback rate is 3 frames/s.
https://elifesciences.org/articles/84396/figures#video9

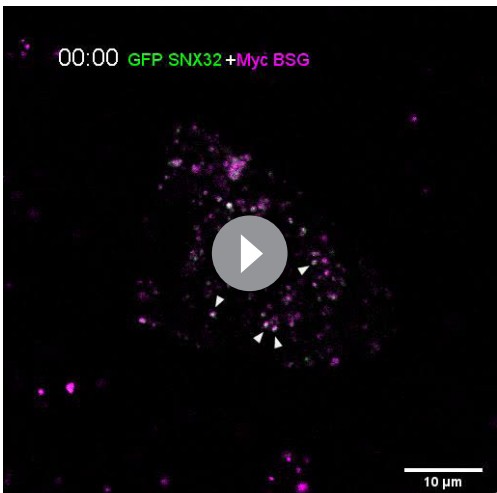

**Video 10.** SNX32 co-traffic with BSG. Neuro2a cells stably expressing cMyc-BSG (magenta) were co-transfected with plasmid encoding and GFP-SNX32 (green) was processed as detailed in the 'Materials and methods' section. Videos were captured in free run mode, without intervals in Olympus FV3000 confocal laser-scanning microscope at 37°C,5% $CO_2$ with moisture control. ZDC-Z Drift compensation was used to correct focus drift during time courses. Frames were collected every 6.4 s for 4 min 49 s. Playback rate is 3 frames/s. The co-trafficking events are indicated by white arrowheads.

https://elifesciences.org/articles/84396/figures#video10

it itself contributes to cargo recognition through its PX domain (*Figure 6D–G*, *Figure 7N–O*). Besides, it is noted in heterodimeric associations of SNX1/SNX5 and SNX1/SNX6, while SNX5, SNX6 lead cargo recognition (*Simonetti et al., 2019*; *Simonetti et al., 2017*; *Kvainickas et al., 2017*; *Niu et al., 2013*), SNX1, SNX2 are known to contribute to membrane remodeling establishing the mutual cooperativity of the individual SNX proteins during cargo sorting. Also, it has been reported in the case of SNX3-retromer (*Lucas et al., 2016*), SNX27-SNX1/2 (*Yong et al., 2021*) that the interaction among the individual members of the above complexes enhances their affinity to the cargo. Thus, it could be possible that the cargo recognition, membrane recruitment, as well as membrane remodeling by SNX32 could be facilitated by the cooperative mode of interactions.

In this context, the role of motor proteins in driving the cargo containing vesicles from the source compartment to the destination is of particular interest. For instance, in the case of SNX1/5 and SNX1/6 complex, the interaction of SNX5/SNX6 with p150glued, an activator of motor protein Dynein, is necessary for the tubular sorting carrier formation (*Hong et al., 2009*; *Wassmer et al., 2009*). Further, the affinity difference between the interaction with the motor and the PIns in the destination membrane commission the cargo dislodging, as delineated in the case of SNX1/SNX6 complex (*Niu et al., 2013*; *Hong et al., 2009*). Since SNX32 belongs to the SNX5/SNX6 cluster, the possibility of SNX32 carrying out the cargo sorting in a similar course is highly probable. In addition, the investigation of CIMPR/TfR trafficking was focused on understanding the contribution of SNX32 in retrograde/recycling trafficking route (*Figure 10*). We showed that the role of SNX32 in TfR fits well into the already existing model of the transferrin trafficking route which includes SNX4 and Rab11 (*Figure 5A–G*).

In our study, we have observed that SNX32 is able to interact with multiple SNX-BAR family members (*Figure 1A and B*) as well as with multiple cargoes (*Figure 6A–C*, *Figure 7N and O*). In addition to SNX1/SNX4, SNX8, the other SNX-BAR family member, could also interact with SNX32 (*Figure 1A and B*). Keeping in account SNX8's role in early endosome to TGN transport (*Dyve et al., 2009*), it will be interesting to explore whether/how these two SNXs pair-up to facilitate cargo sorting. In our study, we established TfR, CIMPR, and BSG as intracellular cargoes for SNX32. Thus, it provides a means to transport cargo from early endosomes to plasma membrane via recycling endosomes/TGN in anterograde/retrograde fashion, respectively (*Figure 10*). Further, we were able to demonstrate that F131 of SNX32 is a crucial amino acid in mediating the interaction with the cargoes (*Figure 6I–K*, *Figure 7P–Q*). Though the mechanistic insight into the sorting events is still enigmatic for the individual cargo molecules, it is likely that the drill would be similar to that observed for β2-adrenergic receptor or Wntless (*Varandas et al., 2016*) where the sorting is ultimately limited/decided by the relative concentration of the cargoes in membrane subdomains (*Simonetti et al., 2019*).

The relatively higher expression of SNX32 in brain tissue and its absence in non-higher metazoans suggests the specific necessity of the protein in the context of neuronal differentiation. Further, it is interesting to note that the interactome of SNX32 encompasses proteins such as PGRMC1 (*Bali et al., 2013*), ROBO1 (*Andrews et al., 2006*; *Bagri et al., 2002*; *Fouquet et al., 2007*; *Kidd et al., 1998*), SV2a (*Janz et al., 1999*), and BSG, which are known to be vital in neuronal development (*Simonetti*

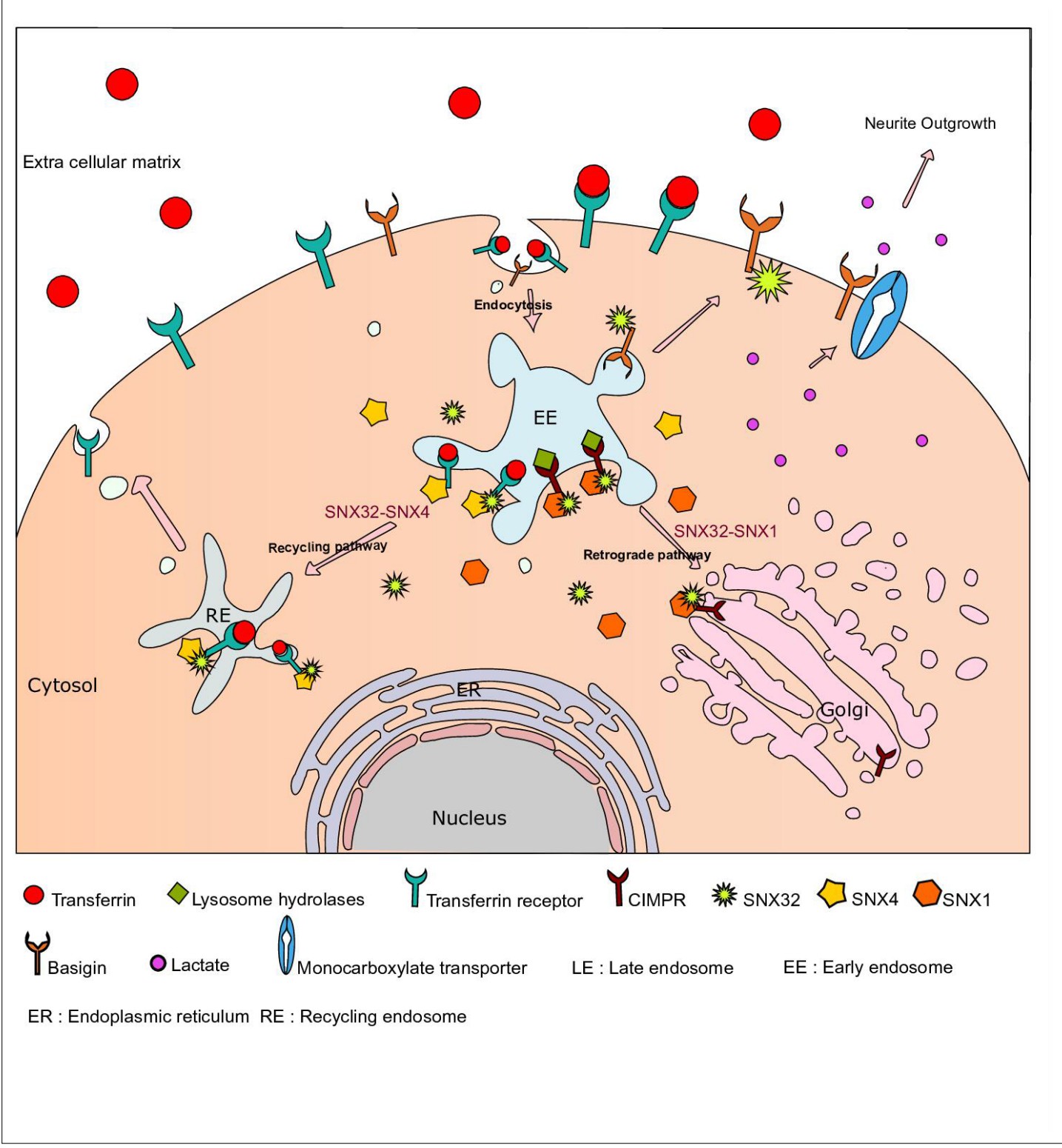

**Figure 10.** Graphical abstract showing SNX32's multifaceted role in trafficking and neuroglial coordination. SNX32 through its BAR domain associate with SNX4, co-traffic TfR from early endosome to recycling endosomes. Similarly, SNX32 interacts with SNX1, co-traffic CIMPR from early endosome to TGN. Further, the proposed role of SNX32 contributing to the endosome to surface trafficking of BSG. BSG being the co-chaperone of monocarboxylate transporter (MCT) 20, 26, 54 helps in the surface localization of MCTs. In conclusion, the derailing of SNX32-mediated BSG trafficking reduces the surface population of BSG as well as MCTs, disrupting the neuroglial coordination and manifests as neurite differentiation defect in cellulo. The model depicts the diverse cargo trafficking route in which SNX32 functions, and a glimpse of complexity introduced by means of its ability to participate into distinct protein complexes.

*et al., 2017*). This study begins to delineate a role for SNX32 in regulating neuronal differentiation through its cargo protein, BSG (*Figure 10*). BSG contributes to a plethora of physiological functions, including sensory and nervous system functioning (*Muramatsu, 2016*; *Igakura et al., 1998*; *Philp et al., 2003*). BSG's interaction with integrin is necessary for cytoskeletal rearrangements (*Curtin et al., 2005*), which is crucial during neurite outgrowth. Our data suggest that SNX32 regulates the surface localization of BSG (*Figure 9D–G*) via its role in plasma membrane recycling of the latter from endosomes, explaining the observed phenocopying of SNX32 and BSG. Further, it has been shown that BSG acts as a chaperone for the MCTs, which plays a crucial role in lactate shuttling and thereby contributes to energy metabolism in neurons. Since lactate accounts for a significant share of energy sources in neurons, the reduced lactate shuttle consequently contributes to defects in neuronal development (*Jha and Morrison, 2020*; *Rossi et al., 2023*). Thus, SNX32, via its role in cell surface transport of BSG, may also regulate MCTs and consequently contribute to energy metabolism in developing neurons (*Figure 8A–L*). Moreover, the enrichment of kinesin-1 heavy chain (KiF5b) in the differential proteomics data of SNX32 suggests that this motor protein may be a potential interactor of SNX32 in neuronal cells. Being a plus-end-directed motor, KiF5 plays an essential role in cargo transport during neurite extension, and accordingly, inhibition of KiF5 abrogates axon specification (*Matsuzaki et al., 2011*). Taken together, it is tempting to speculate that the association with KiF5 may enable SNX32 to contribute to the long-range trafficking of proteins like BSG to the surface.

# Materials and methods
## Plasmids and antibodies
pEGFP C1 SNX32 FL, pEGFP C1 SNX1, pEGFP C1 SNX4, mCherry C1 SNX1 were a kind gift from Prof. Peter J Cullen. pGEX6P1 GFP Nanobody (#61838, Addgene) was purchased from Addgene and subcloned to pGEX6P1 for improved induction, pGEX6P1mCherry nanobody (#70696, Addgene) was purchased from Addgene. pLVX TRE 3G and pLVX EF1α Tet3G were part of Tet-On 3G Inducible Expression Systems (631363, Takara Bio Inc). pCMV6-Entry Myc DDK CD147 (BSG) (#: RC203894 Origene). pLVX TRE3G pHluorin BSG (subcloned from #RC203894), pLVX-TRE3G GFP-SNX32#4r were synthesized by commercial cloning service provider GenScript USA Inc pmCherry-Rab 11 was a kind gift from Prof. Marino Zerial. pIRES neo2 CD8 CIMPR was a kind gift from Prof. Matthew NJ Seaman. pmCherry C2-PH $^{PLC\delta}$ was a kind gift from Prof. Pietro De Camilli. pEGFP C1 PH$^{OSBP}$ was a kind gift from Prof. Tamas Balla. pLVX TRE3G GFP-SNX32FL, pcDNA3 HA N(I)-SNX32 FL, pmCherry C2-SNX32 FL, pcDNA3 HA N(I)-SNX 32 ΔN, pcDNA3 HA N(I)-SNX32 ΔC, His-SNX32FL, His-SNX32ΔN, His-SNX32ΔC, GST-SNX32FL, GST-SNX32ΔC were subcloned from pEGFP C1-SNX32FL, RFP C1-SNX4 was subcloned from pEGFP C1-SNX4. pcDNA3 HA N(I)-SNX32#4r, pcDNA3 HA N(I)-SNX32#4rΔN, pEGFP-SNX4Y258E, pEGFP-SNX4S448R, pcDNA3 HA N(I)-SNX32 R220E, pcDNA3 HA N(I)-SNX32A226E, pcDNA3 HA N(I)-SNX32E256R, pcDNA3 HA N(I)-SNX32 Q259R, pcDNA3 HA N(I)-SNX32 R366E were constructed using site-directed mutagenesis.

Antibodies used in this study are anti-GFP (11814460001, Roche; WB,1:3000), anti-HA(C29F4, Cell Signaling Technology, WB, 1:1000, IF, 1:500), anti-His (MA1-21315, Invitrogen; WB, 1:10,000), anti-EEA1 antibody was a kind gift from Prof. Marino Zerial, anti-HA (sc7392, Santa Cruz, WB/IF, 1:500), anti-TGN46 (AHP1586, AbD Serotec, IF, 1:200), anti-transferrin receptor (13-6800, Invitrogen; WB, 1:1000, IF, 1:200), transferrin 488 (T13342, Invitrogen, IF, 5μg/ml), transferrin 568 (T23365, Invitrogen, IF, 5μg/ml), transferrin 647 (T23366 Invitrogen, live imaging, 10 μg/ml), anti-SNX1 (611482, BD Biosciences, IF, 1:500) anti-Vinculin (V9131, Sigma-Aldrich, WB, 1:1000), anti-CIMPR (ab2733, Abcam, IF, 1:500), anti-CD8 (153-020, Ancell, IF, 1:500, live imaging, 1:300), anti-CIMPR (ab124767, Abcam, WB, 1:50,000), anti-mCherry (M11217, Invitrogen; WB, 1:1000, IF, 1:500), anti-CD147(345600, Invitrogen; WB/IF, 2 μg/ml), c-Myc (9E10-sc40, Santa Cruz, IF, 1:500), and anti-ARF6 antibody was a kind gift from Dr. Vimlesh Kumar.

HeLa cells were a kind gift from Prof. Marino Zerial, Max Planck Institute of Molecular Cell Biology and Genetics, Dresden, and also procured from ATCC (ATCC no.: CCL-2TM, lot number: 70046455). HEK-293T cells were procured from ATCC (ATCC no.: CCL-2TM, lot number: 70046455). U87MG and Neuro2a cells were acquired from Cell Repository, National Centre for Cell Science Pune, India. All the cell lines have been authenticated using STR profiling except Neuro2a. They were also tested negative for mycoplasma contamination.

## Uptake media composition

For HeLa-DMEM containing 10% heat-inactivated South American FBS, 1× penicillin/streptomycin, 20 mM HEPES (pH 7.5).

For Neuro2a-MEM containing 10% heat-inactivated South American FBS, 1× pencillin/streptomycin, 2 mM sodium pyruvate, 20 mM HEPES (pH 7.5).

## Cleared cell lysate preparation

Cells were lysed with lysis buffer (50 mM Tris [pH 8], 150 mM NaCl, 0.5% NP40) for 5 min on rotamer. Later, the lysates were cleared by centrifugation at 19,000 × *g* for 20 min at 4°C. The supernatant fraction was collected without disturbing the pellet and used for experiments.

## siRNA transfection

All ON-TARGET plus siRNA SMART pools were purchased from Dharmacon. HeLa cells were transfected with SMART pool siRNAs against negative control siRNA-scrambled (D-001810-10), SNX32 (L-017082-01), SNX6 (L-017557-00) (sequence information is included in *Supplementary file 1*). Cells were transfected with 15–20 nm siRNA using DharmaFECT 1 (Dharmacon) following the manufacturer's protocol and incubated for a maximum of 72 hr before any further analysis.

## shRNA transfection

All shRNA clones (sequence information is included in *Supplementary file 1*) were part of the MISSION shRNA product line from Sigma-Aldrich. The TRC1.5 pLKO.1-puro non-mammalian shRNA Control Plasmid DNA (SHC002) is a negative control containing a sequence that should not target any known mammalian genes but engage with RISC. The SNX32 clones – shSNX32#4 (TRCN0000181072) and shSNX32#6 (TRCN0000180862) – and the BSG clones – shBSG#7 (TRCN0000006733) and shBSG#8 (TRCN0000006734) – were screened for maximum knockdown efficiency compared to other available clones targeting the same protein. The cells were transfected with shRNA clones (as mentioned in the figures and legends) using Lipofectamine LTX with Plus Reagent (Invitrogen) for a maximum of 48 hr before any further analysis.

## RNA extraction and quantitative real-time PCR

Total RNA was extracted from the cells using RNA easy kit (QIAGEN, Cat# 74104), and cDNA was prepared using the High-Capacity RNA-to-cDNA kit (Life Technologies, Cat# 4387406). Real-time qPCR reactions were performed using the SYBR Green Kit and corresponding primers (sequence information is included in *Supplementary file 1*) on Applied Biosystems 7300 Real-Time PCR System or Thermo Quant Studio 3.0.

## shRNA-resistant SNX32

pLVX TRE 3G SNX32 resistant (shSNX32#4r) to shRNA clone – shSNX32#4 (TRCN0000181072) – was synthesized by commercial cloning service provider GenScript USA Inc, which was further subcloned into pcDNA3 HA vector backbone. pcDNA3 SNX32ΔN resistant (shSNX32ΔN#4r) was constructed following site-directed mutagenesis using the primer 5'TTCGAACACGAACGGACAT 3'.

## GFP/mCherry nanobody-mediated immunoprecipitation (GBP/mCBP-IP)

GFP binding protein (GBP) (*Fridy et al., 2014*)/mCherry binding protein (mCBP), also called GFP/mCherry-nanobody tagged with GST, is a single-chain VHH antibody domain developed with specific binding activity against GFP/mCherry protein. 20 µg of GST-GFP/mCherry nanobody (GFP/mCherry binding protein) was allowed to bind to glutathione sepharose beads in 1× PBS for 1 hr at 4°C. Followed by removal of unbound proteins, the beads were incubated with cell lysate containing overexpressed GFP /mCherry empty vector, or GFP/mCherry-tagged target protein for 2 hr at 4°C. After incubation, unbound protein residues were removed by washing the beads-nanobody-Target protein complex thrice with 400 µl 1× PBS. Further, samples were prepared, resolved on SDS-glycine gel, and analyzed after immunoblotting. 5% of the cleared cell lysate was used as input.

## Dimeric structure prediction

Predictions of dimeric models were performed using AlphaFold2-Multimer (ColabFold). For further analysis, top-ranked models with the best prediction quality were selected. Analysis of the structures

and generation of molecular graphics images was carried out in PyMOL (version 2.5.2; Schrödinger). Buried surface area calculations were performed using the PDBsum webserver (*Jumper et al., 2021*; *Evans et al., 2021*). The amino acid sequences corresponding to the BAR domain of SNX1 (NCBI RefSeq: NM_003099.5), SNX4 (NCBI RefSeq: NM_003794.4), and SNX32 (NCBI RefSeq: NP_689973.2) with residue numbers 303–522, 205–448, and 180–400, respectively, were used as input. Top-ranked models with the best prediction quality (higher overall pLDDT score) were selected for further analysis. The obtained models were analyzed for intermolecular polar interactions (hydrogen bonds and salt bridges) to determine potential key residues participating in the dimer formation. Analysis of the structures and generation of molecular graphics images was carried out in PyMOL (version 2.5.2; Schrödinger). Buried surface area calculations of the dimeric interface were performed using the PDBsum webserver (ref: PMID: 28875543).

## Site-directed mutagenesis

Mutagenesis PCR was used to introduce mutations into SNX4 and SNX32. A mutant construct was generated by PCR using GFP-tagged SNX4 or HA-tagged SNX32 as template DNA, with an appropriate forward primer and a reverse primer that introduced the mutation. The PCR product was treated with Dpn1 to digest the template and hybrid DNA, followed by transformation to DH5αcells. The constructs were confirmed by DNA sequencing.

## Indirect immunofluorescence

Cells were fixed using 4% PFA or 100% methanol (MeOH, Sigma-Aldrich) for 10 min (PFA) or 15 min (MeOH) at room temperature (PFA) or –20°C (MeOH). Following PFA fixation, the cells were permeabilized using 0.1% Triton X-100 (Sigma-Aldrich) in PBS. Cells fixed using MeOH were directly blocked using 5% fetal bovine serum in PBS before incubating with primary and secondary Alexa-labeled antibodies. The coverslips (1943-10012A, Bellco) were mounted using Mowiol on glass slides and imaged using Zeiss LSM 780 laser-scanning confocal microscope ×63/1.4 NA oil immersion objective lens or Olympus FV3000 confocal laser-scanning microscope with a ×60 Plan Apo N objective (oil, 1.42 NA). Data from three independent experiments were subjected to analysis by the automated image analysis program, Motion Tracking (*Kalaidzidis, 2007*; *Rink et al., 2005*; http://motiontracking.mpi-cbg.de) as described in detail *Source data 2*.

## Phenyl arsenide oxide/Wortmannin treatment

Phenyl arsenide oxide/Wortmannin treatment was followed as previously reported (*Hong et al., 2009*). Briefly, HeLa cells were transiently transfected with individual target protein plasmids and incubated for 12–14 hr at 37°C, 5% $CO_2$. Further, the cells were treated with PAO (15 μM)/Wortmannin (200 nM) in uptake medium for 15 min at 37°C, proceeded for immunofluorescence.

For the in vitro membrane relocalization assay, following the PAO treatment, cells were used for membrane fractionation assay and equal amount of protein was resolved using SDS-PAGE and analyzed after immunoblotting.

## Generation of stable cell lines

HeLa cells stably expressing GFP-SNX32 from an inducible promoter were established following the distributor protocol (Takara Bio Inc). GFP-SNX32 full-length gene was cloned onto pLVX-TRE3G Vector. For viral titer preparation, 7 μg of pLVX-TRE3G GFP-SNX32 or pLVX-pEF1a-Tet3G Vector plasmid DNA was diluted in sterile water to a final volume of 600 μl. Further, the diluted DNA was added to Lenti-X Packaging Single Shots for 10 min at room temperature to allow nanoparticle complexes to form. Followed by incubation, the entire 600 μl of nanoparticle complex solution was added to 80% confluent HEK-293T cells. The cells were incubated at 37°C, 5% $CO_2$ for 24–48 hr. Viral titers were harvested by centrifuging briefly (500 × *g* for 10 min) followed by filtering through a 0.45 μm filter to remove cellular debris. Later, HeLa cells were transduced with the viral titers of pLVX-TRE3G GFP-SNX32 and pLVX-pEF1a-Tet3G mixed in 1:1 ratio and topped up with DMEM complete media containing 4 μg/ml concentration of Polybrene. 24 hr past transduction, the culture media was removed and replaced with complete DMEM containing G418 (600 μg/ml) and puromycin (10 μg/ml) selection media. The cells were allowed to divide and form colonies for 14 d before passaging. Cells were maintained in complete DMEM containing G418 (400 μg/ml) and puromycin (0.25 μg/ml).

Neuro2a cells stably expressing pHluorin BSG from an inducible promoter were established by co-transfecting pLVX-TRE3G pHluorin BSG, pLVX-pEF1a-Tet3G Vector plasmid DNA in 1:1 ratio using Lipofectamine LTX with Plus Reagent (Invitrogen). 24 hr post-transfection, the culture media was replaced with complete MEM containing G418 (600 µg/ml) and puromycin (10 µg/ml) selection media. The cells were allowed to divide and form colonies for 18 d before passaging. Cells were maintained in complete MEM containing G418 (400 µg/ml) and puromycin (0.25 µg/ml).

## Lipid overlay assay

Lipid overlay assay was performed using PIP strips (nitrocellulose membrane spotted with eight phosphoinositides and seven other biologically relevant lipids) according to the manufacturer's instructions (Thermo Scientific). Briefly, the membrane was incubated with 0.7 mg/ml His SNX32ΔC diluted in blocking buffer (TBS-T + 3% BSA) for 13 hr at 4°C with gentle mixing. The His SNX32ΔC protein bound to the lipids was detected by anti-His antibody (1:10,000; Thermo Scientific).

## Super-resolution microscopy sample preparation

As mentioned earlier, HeLa cells expressing the protein of interest were fixed in 4% PFA for 10 min at room temperature and proceeded for indirect immunofluorescence. Briefly, the cells were permeabilized with 0.1% Triton X-100 for 12 min at room temperature and immunostained using GFP, mCherry primary antibodies, followed by Alexa-labeled secondary antibodies. The coverslips were mounted on glass slides using mounting media without DAPI. Image was captured using Nikon N-SIM S system in 3D-SIM mode (sequentially) with laser wavelengths 488 nm and 561 nm. For each Z plane and each wavelength, 15 images were captured (three different angles and five different phases). Images were captured in Nikon N-SIM S demo system and reconstructed using Nikon software NIS Elements version 5.30.

## Membrane fractionation and enrichment assay

The assay was performed as previously reported (*Seaman et al., 2009*). Briefly, HeLa cells grown in T25 tissue culture flasks were washed twice with PBS and drained completely. Later, the cells were snap frozen using liquid nitrogen and then quickly thawed at room temperature. Cells were scraped off in 0.5 ml of lysis buffer (0.1 M Mes-NaOH pH 6.5, 1 mM magnesium acetate, 0.5 mM EGTA, 0.2 M sucrose, and 1xPIC). The pellet (which contained the membrane proteins) was separated from the supernatant (which contained the cytosolic proteins) by centrifuging at 10,000 × *g* for 10 min. The pellet was then solubilized in 0.3 ml of lysis buffer (50 mM Tris-HCl, pH 7.4, 150 mM NaCl, 1 mM EDTA, 1% Triton X-100, 0.1% SDS) before next round of centrifugation at 10,000 × *g* for 10 min. The supernatant from the second spin now contained membrane and membrane-associated proteins, which was used as membrane-enriched fraction for in vitro pulldown assay. Equal amount of pellet and supernatant was loaded for analyzing membrane relocalization assay in the presence of PAO.

## Transferrin pulse-chase assay

This was followed as previously reported (*Magadán et al., 2006*). Briefly, HeLa cells were transfected with scrambled/individual siRNA SMART pool or SHC002/shSNX32 clones for 70 hr (siRNA) or 46 hr (shRNA) followed by serum starvation for 2 hr to deplete the endogenous transferrin population. Cells were then washed with uptake media and incubated with 5 µg/ml transferrin (Alexa Fluor 488/568 conjugated) at 37°C for 30 min. After completing the pulse period, the cells were washed and proceeded for chase using unlabeled Holotransferrin 100 µg/ml. Cells were fixed at specified periods and proceeded for immunofluorescence.

## CD8 uptake assay

HeLa cells stably expressing GFP-Golph3 and CD8-CIMPR were transfected with scrambled/individual siRNA SMART pool or SHC002/shSNX32 clones for 72 hr (siRNA) or 48 hr (shRNA). Further, cells were incubated with anti-CD8 monoclonal antibody for 60 min on ice, followed by two quick washes with uptake media to remove unbound antibodies. Chase was done by incubating the cells in pre-warmed uptake media at 37°C for 30 min followed by fixation and proceeded for immunofluorescence.

## Protein purification

### Purification of His-SNX32ΔC/His-SNX12

*Escherichia coli* BL21 (DE3) cells were transformed with plasmids encoding His SNX32ΔC. Colonies were screened, and culture induction conditions were standardized. The culture was grown at 37 °C until $OD_{600}$ reached ~0.6–0.8. Temperature was then lowered to 16 °C, and protein expression was induced by adding 0.1 mM isopropyl β-d-1-thiogalactopyranoside (IPTG). After ~15 hr, cells were harvested and homogenized in lysis buffer (20 mM Tris, pH 8.0, 400 mM NaCl, 2 mM 2-mercaptoethanol [βME], 10 mM Imidazole and 1 mM PMSF). Cells were lysed by sonication for 2 min and subjected to high-speed centrifugation to remove insoluble debris. The supernatant was then mixed with Ni-NTA beads for 20 min. Unbound proteins were washed off with wash buffer (20 mM Tris [pH 8.0], 200 mM NaCl, 2 mM βME, 10 mM imidazole). Protein was eluted in buffer containing 250 mM imidazole. The eluted protein was subjected to buffer exchange (20 mM Tris [pH 8.0], 200 mM NaCl, 2 mM 2-mercaptoethanol [βME], 10% glycerol) in Jumbosep Centrifugal Devices and flash frozen to store at –80°C.

### Co-purification of GST SNX1/His SNX32ΔN

The procedure was followed as reported by *Yong et al., 2018*. Briefly, plasmids encoding GSTSNX1 and His SNX32ΔN were co-transformed into *E. coli* BL21 (DE3) strain, grown in Luria-Bertani (LB) agar plates supplied with 50 μg/ml ampicillin and 30 μg/ml kanamycin. An overnight liquid culture of 10 ml was used to initiate a 1 l expression of SNX1/SNX32ΔN complex. The culture was grown at 37 °C until $OD_{600}$ reached ~0.6–0.8; later the temperature was lowered to 22 °C, and protein expression was then induced by adding 0.5 mM IPTG. After 14–15 hr, cells were harvested and homogenized in lysis buffer (20 mM Tris, pH 8.0, 200 mM NaCl, 2 mM 2-mercaptoethanol [βME], 10 mM imidazole, 10% [v/v] glycerol, and 1 mM PMSF). Cells were lysed by sonication for 2 min (10 s ON–10 s OFF pulses) and subjected to high-speed centrifugation to remove insoluble debris. The supernatant was then mixed with Ni-NTA beads for 20 min. Unbound proteins were washed off with wash buffer (20 mM Tris, pH 8.0, 200 mM NaCl, 2 mM βME, 10 mM imidazole). Fractions were concentrated and flash frozen before storing at −80 °C.

## Liposome preparation

All lipids were purchased from Avanti Polar Lipids. The fluorescent crosslinker lipid BODIPY-diazirine PE was synthesized as described earlier (*Jose et al., 2020*; *Jose and Pucadyil, 2020*). Lipids in the required proportion were aliquoted from chloroform stocks into a glass tube and dried under high vacuum for an hour. Dried lipids were hydrated in deionized water at 50°C for 1 hr and then vortexed to generate multilamellar vesicles, which were then extruded through 100 nm polycarbonate filters Whatman (GE Healthcare, IL). Liposome stocks were prepared at a concentration of 0.5 mM. Liposomes contained phosphoinositide at 5 mol% and BODIPY-diazirine PE at 1 mol%.

## PLiMAP assay

PLiMAP assays were carried out as described previously (*Jose et al., 2020*; *Jose and Pucadyil, 2020*), with a few modifications. Proteins and liposomes were mixed at a final concentration of 2 and 200 μM, respectively, in a total volume of 30 μl of 20 mM HEPES (pH 7.4) with 150 mM NaCl buffer. The reaction mix was transferred to a 96-well plate and incubated for 30 min at 25°C in the dark. The plate was then exposed to 365 nm UV light (UVP crosslinker CL-1000L) and an energy setting of 200 mJ.$cm^{-2}$ for 1 min. The plate was placed at a distance of 3 cm from the lamp. Sample buffer was then added to the reaction mixture, aliquoted out into an Eppendorf, boiled at 99°C, and resolved using SDS-PAGE. Gels were first scanned for BODIPY fluorescence on an Amersham Typhoon Bimolecular Imager (Cytiva Lifesciences) and later fixed and stained with Coomassie Brilliant Blue.

## SILAC methodology

For SILAC, SH-SY-5Y cells stably expressing GFP or a GFP-tagged construct of the protein of interest were cultured for at least six doublings in SILAC DMEM (89985; Thermo Fisher Scientific) supplemented with 10% dialyzed FBS (F0392; Sigma-Aldrich). Cells expressing GFP were grown in media containing light amino acids (R0K0), whereas cells expressing the GFP-tagged protein of interest were grown in medium (R10K8 or R6K4). Amino acids R10,R6, R0, and K0 were obtained from

Sigma-Aldrich, whereas K4 was from Thermo Fisher Scientific. Cells where lysed in immunoprecipitation buffer (50 mM Tris-HCl, 0.5% NP-40, and Roche protease inhibitor cocktail) and subjected to GFP trap (ChromoTek). Precipitates were pooled and separated on NuPAGE 4–12% precast gels (Invitrogen) before liquid chromatography–tandem mass spectrometry analysis on an Orbitrap Velos mass spectrometer (Thermo Fisher Scientific).

## In vitro pulldown assay

20 µg of His SNX32ΔC was allowed to bind to Ni-NTA beads for 20 min, 4°C with end-to-end mixing. The unbound protein was washed off with PBS, followed by incubation with membrane-enriched HeLa cell lysate for 2 hr, 4°C with end-to-end mixing. The samples were resolved using SDS-PAGE and analyzed after immunoblotting.

## Neurite outgrowth assay

Neurite outgrowth assay was performed as previously reported (*Ma et al., 2015*). Briefly, Neuro2a cells were transfected with scrambled or individual siRNA SMART pools or shc002/shSNX32/shBSG clones. Following 24 hr (siRNA) or 6 hr (shRNA) of transfection, the medium was replaced with MEM containing 1% fetal bovine serum supplemented with 10 µmol/l retinoic acid (RA) for another 48 hr to induce neurite outgrowth. The formation of neurites was observed using Axio Vert.A1 Inverted Transmitted Light Microscope (Carl Zeiss Microscopy GmbH, Göttingen, Germany) after cell fixation or live images were captured using JuLIBr inverted microscope (NanoEnTek). The JuLIBr system is equipped with a station unit that runs inside a $CO_2$-regulated incubator and a scope unit that runs outside the incubator. Phase-contrast images were captured every 1 min for 48 hr using a ×4 objective and a CMOS camera with a pixel length of 0.586 µm. The cells with neurites were counted using ImageJ software.

## Rescue experiments

### Transferrin pulse-chase

HeLa cells were transiently transfected with SNX32 shRNA clone shSNX32 #4 to deplete the expression of SNX32 for 34 hr. Further, the cells were transfected with shSNX32 #4-resistant HA-tagged SNX32 for 12 hr; following that, the cells were serum-starved for 2 hr. Transferrin pulse-chase was carried out. Cells were fixed at specified periods and proceeded for immunofluorescence.

### Neurite outgrowth

Neuro2a cells stably expressing doxycycline-inducible pLVX SNX32#4-resistant construct were transiently transfected with shSNX32#4 to deplete the expression of SNX32 for 48 hr (until the completion of the experiment). Following 6 hr after transfection, the cells were induced with doxycycline for expressing shSNX32 #4-resistant SNX32 and RA in reduced serum-containing media. The formation of neurites was observed using Axio Vert.A1 Inverted Transmitted Light Microscope (Carl Zeiss Microscopy GmbH) after cell fixation or live images were captured using JuLIBr inverted microscope (NanoEnTek). The JuLIBr system is equipped with a station unit that runs inside a $CO_2$-regulated incubator and a scope unit that runs outside the incubator. Phase-contrast images were captured every 1 min for 48 hr using a ×4 objective and a CMOS camera with a pixel length of 0.586 µm. The cells with neurites were counted using ImageJ software.

## Confocal live-cell microscopy

HeLa cells seeded on glass-bottom dishes were transfected with respective constructs using Lipofectamine LTX with Plus reagent (Invitrogen) and incubated for 12 hr, 37°C, 5% $CO_2$. Images were acquired using Olympus FV3000 confocal laser-scanning microscope for *Videos 1 and 2*.

To capture transferrin co-traffic, HeLa cells were transiently transfected with mCherry-SNX32 and GFP-SNX4. Later cells were serum-starved for 2 hr and incubated with Transferrin (Alexa Fluor 647 conjugated) in uptake media, for 2 min prior to imaging. Videos were captured for 5 min, without intervals in Olympus FV3000 confocal laser-scanning microscope with a ×60 Plan Apo N objective (oil, 1.42 NA) on an inverted stage. Images were acquired and processed using FV31S-SW software and ImageJ software, respectively.

To capture CD8-CIMPR co-traffic, HeLa cells stably expressing pLVX TRE3G GFP-SNX32 were transiently transfected with mCherry-SNX1 for 4 hr followed by doxycycline induction for 12 hr. Later, cells were treated with anti-CD8 antibody and incubated for 1 hr at 4°C. Unbound antibody was washed using uptake media followed by incubation with Alexa 647-labeled secondary antibody. Cells were washed twice with uptake media prior to imaging. Videos were captured for 5 min, without intervals in Olympus FV3000 confocal laser-scanning microscope with a ×60 Plan Apo N objective (oil, 1.42 NA) on an inverted stage. Images were acquired and processed using FV31S-SW software and ImageJ software, respectively.

To capture GFP-SNX32 and Myc-BSG co-trafficking, Neuro2a cells were transiently transfected with pLVX TRE3g GFP-SNX32 and Myc-BSG for 4 hr followed by doxycycline induction for 12 hr. Later, cells were treated with anti-Myc antibody and incubated for 1 hr in 4°C. Unbound antibody was washed using uptake media followed by incubation with Alexa 647-labeled secondary antibody. Cells were washed twice with uptake media prior to imaging. Videos were captured for 5 min, without intervals in Olympus FV3000 confocal laser-scanning microscope with a ×60 Plan Apo N objective (oil, 1.42 NA) on an inverted stage. Images were acquired and processed using FV31S-SW software and ImageJ software, respectively.

## Lactate quantification assay

Lactate quantification assay was performed following the manufacturer's protocol (MAK064, Sigma-Aldrich). Briefly, U87MG cells were plated on a 96-well plate (1000 cells per cell well) and cultured overnight in MEM containing 10% FBS, followed by treatment with scrambled/SNX6/SNX32 siRNA SMART pools or SHC002/shSNX32/shBSG clones. After 72 hr (siRNA) or 48 hr (shRNA) of transfection, the supernatant medium was collected and used for lactate quantification.

## TIRF microscopy

TIRF microscopy was performed using a Nikon Eclipse Ti2 microscope, equipped with an incubation chamber (37°C, 5% $CO_2$), a ×60 TIRF objective (oil-immersion, Nikon), an sCMOS camera (Neo, Andor), a 100 W mercury lamp (C-LHG1 Mercury). Stable Neuro2a cells expressing pHluorin BSG were seeded on a glass-bottom dish (100350, SPL), followed by treatment with scrambled/SNX6/SNX32 siRNA SMART pools or shc002/shSNX32 clones for a maximum of 72 hr (siRNA) or 48 hr (shRNA). 12 hr prior to imaging, cells were transfected with doxycycline for pHluorin BSG induction. Data from three independent experiments were subjected to analysis by the automated image analysis program, Motion Tracking (*Kalaidzidis, 2007*; *Rink et al., 2005*; http://motiontracking.mpi-cbg.de). The imaging frames were randomly selected, and a minimum of five videos of 1 min duration, without interval, were acquired for each experimental condition in a given setup. The objects were identified based on their size, fluorescence intensity, and other parameters by Motion Tracking software. The number of objects detected was normalized with the size of the vesicles and averaged with the number of cells per frame were plotted and compared in GraphPad Prism 9.

## Acknowledgements

We extend our gratitude to Prof. Peter J Cullen and Dr. Boris Simonetti (University of Bristol, UK) for their fruitful discussions, guidance, and for providing the SILAC data. We also acknowledge Ms. Katy (Prof. Cullen's group, University of Bristol) for resource sharing. We sincerely thank Prof. Matthew NJ Seaman (Cambridge Institute for Medical Research) for his critical comments on the study and also for resource sharing. We thank Prof. Marino Zerial (Max Planck Institute of Molecular Cell Biology and Genetics), Prof. Pietro De Camilli (Yale's Boyer Center for Molecular Medicine), and Prof. Tamas Balla (National Institutes of Health) for sharing plasmids for mammalian expression. We express gratitude to Dr. Joseph Ong, UCLA, for assisting in the gene designing of shRNA-resistant SNX32. Dr. Ganesh and Pankaj (Olympus) for their unconditional support in image acquisition. We acknowledge the FIST facility at IISER Bhopal by DST for providing confocal microscopy facilities. We are grateful to Dr. Vimlesh Kumar (IISER Bhopal) for his suggestions and for sharing the ARF6 antibody. We sincerely thank Dr. Raghuvir Singh Tomar, Dr. Vikas Jain, and Dr. Himanshu Kumar for providing access to various instruments. We thank Prabal Kumar Chakraborty and Suparno Gupta of Towa Optics (I) Pvt. Ltd. for assistance in image acquisition using the Nikon N-SIM S demo system. We acknowledge Dr. Manish Kumar Dwivedi (IISER Bhopal), Ms. Shikha Kushwaha (IISER Bhopal), Mr. Sajeev TK (IISER

Bhopal), and Mr. Satyam Sharma (IISER Bhopal) for suggestions and discussions. We also thank Rabiya Naaz for lab management.

## Additional information

### Funding

| Funder | Grant reference number | Author |
|---|---|---|
| Science and Engineering Research Board | CRG/2019/004580 | Sunando Datta |
| Department of Biotechnology, Ministry of Science and Technology, India | DBT-JRF | Jini Sugatha |
| Indian Institute of Science | Apr2019/709/BS/iiserb/ Sunando Datta | Sunando Datta |

The funders had no role in study design, data collection and interpretation, or the decision to submit the work for publication.

### Author contributions

Jini Sugatha, Conceptualization, Formal analysis, Supervision, Funding acquisition, Validation, Investigation, Visualization, Methodology, Writing – original draft, Writing – review and editing; Amulya Priya, Formal analysis, Investigation, Visualization, Methodology, Writing – original draft; Prateek Raj, Software, Formal analysis, Methodology, Writing – original draft, Writing – review and editing; Ebsy Jaimon, Formal analysis, Supervision, Investigation, Methodology, Writing – review and editing; Uma Swaminathan, Investigation, Visualization, Methodology, Writing – original draft; Anju Jose, Investigation; Thomas John Pucadyil, Data curation, Supervision, Methodology, Project administration; Sunando Datta, Conceptualization, Resources, Supervision, Funding acquisition, Methodology, Writing – original draft, Project administration, Writing – review and editing

### Author ORCIDs

Jini Sugatha http://orcid.org/0000-0002-5628-4597
Amulya Priya http://orcid.org/0000-0001-8673-5371
Ebsy Jaimon http://orcid.org/0000-0001-6845-2095
Sunando Datta http://orcid.org/0000-0002-1417-0276

### Decision letter and Author response

Decision letter https://doi.org/10.7554/eLife.84396.sa1
Author response https://doi.org/10.7554/eLife.84396.sa2

## Additional files

### Supplementary files

• Supplementary file 1. Sequence details of the primers, siRNA/shRNAs, used in the study. Sheet 1: the primer sequences used to carry out RT PCR, Sheet 2: the sequence details of the shRNA clones used in the study, Sheet 3: the siRNA sequences and product details.

• MDAR checklist

• Source data 1.

• Source data 2.

### Data availability

All data generated or analyzed during this study are included in the manuscript and supporting file; Source Data files have been provided separately.

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
