## [Editor Report]

This manuscript presents a series of important findings about the roles of the BAR-domain containing protein SNX32 in endosomal cargo sorting and in neurite outgrowth. The authors provide compelling evidence for their claims, which will be of interest to those working not only in membrane trafficking but also to cell biologists in general with an interest in neurobiology.

---

## [Decision Letter]

**Decision letter after peer review:**

Thank you for submitting your article "Insights into cargo sorting by SNX32 in neuronal and non-neuronal cells: physiological implications in neurite outgrowth" for consideration by *eLife*. Your article has been reviewed by 3 peer reviewers, including Felix Campelo as the Reviewing Editor and Reviewer #2, and the evaluation has been overseen by Vivek Malhotra as the Senior Editor. The following individuals involved in review of your submission have agreed to reveal their identity: Agata Witkowska (Reviewer #1); Alex J B Kreutzberger (Reviewer #3).

Essential revisions:

1) We all agree that this paper has the potential to become an important paper. However, in its present state, the manuscript is very difficult to follow. The authors should make an effort to rewrite and organize the paper in a manner that helps readers understand and appreciate the data The individual reviews make some suggestions, but the authors should make a substantial effort in improving/rewriting the manuscript, so it flows clearly and the readers are not lost in details.

Regarding the experiments, some simple experiments/clarifications will be required to strengthen the proposal (see individual reports for details):

2) Explain/quantify better what you mean by "colocalization" in the fluorescence microscopy images.

3) Add missing information in the methods section.

4) Provide further evidence of PI3P vs PI4P binding (Reviewers #1 and #2 suggest some experiments along these lines).

5) Discuss if SNX32's main role is largely to direct trafficking steps from early endosomes to recycling endosomes, back to the plasma membrane? And is this a general mechanism for all recycled receptors or specific for TfR, CIMPR, and BSG trafficking? Also, discuss the possibility of cooperative binding with other proteins on endosomes.

*Reviewer #1 (Recommendations for the authors):*

Conclusions of this paper are mostly well supported with experiments; however, some points need additional clarification. I have suggestions and comments that I believe should be addressed before publication.

Comments:

1. Authors state that expression of SNX32 in HeLa cells is comparable to U87MG and Neuro2a although graph in Figure S2A shows reduced by half expression in Neuro2a. This should be clarified in the text. Also, statistical analysis is missing in this graph.

2. In the method description for the colocalization analysis authors state that >35% overlap is considered as colocalization, although in Figure 2B Rab11 or TGN46 are hardly colocalized with SNX32 (less than 20%). However, authors state in the description of the confocal experiment that colocalization was visible (line 187). Authors should rephrase this part to indicate that these results come rather from live-cell microscopy or super-resolution microscopy experiments.

3. Super-resolution microscopy videos (Videos 2,3) seem hard to interpret as colocalization. Here (including Figure 2FG), some additional quantifications would be beneficial. Additionally, since there is quite a substantial difference in colocalization visible by confocal and super-resolution imaging in case of Golgi marker and Rab11, it requires a negative control with another organelle marker that does not show colocalization. Moreover, it is hard for the reader to refer it to the appropriate section in the methods section ("SIM sample preparation"), as the two are not linked by any key word (either "super-resolution" or SIM).

4. Details of all qRT-PCR should be added to the methods sections. Primers used should be listed in the final version of the manuscript.

5. Authors predict heterodimeric BAR structures of SNX32 with SNX1 and SNX4 with AlphaFold2-Multimer; however, details of these analyses are missing from the methods section. Input sequences (or at least residue-ranges) should be stated as well as details of how the obtained models were analyzed that ultimately led to identification of key residues.

6. The PIP-strip experiment (Figure 2H) does not show specificity for PI(3)P or PI(4)P, however authors make a point that it is PI(3)P and PI(4)P that PX of SNX32 binds to and directs to endosomes. This statement in the abstract is not fully justified by experiments. Even though, treatment with PI-inhibitors, shows re-localization from endosomes to the cytoplasm, it needs to be considered that specific subcellular localization could be mediated by interactions with other endosomal partners as well as cooperativity in membrane binding by PX and BAR domains (as suggested by different results with either deltaC or deltaN version of the protein and with very minor effect on the full length). Here, additional experiments looking at least at SNX1 and SNX4 localization upon inhibitor treatment are required as well as more extensive discussion of the results.

7. Authors show that SNX6 and SNX32 have overlapping roles in TfR trafficking; however, do not provide such analysis for CIMPR (for example in Figure S3L).

*Reviewer #2 (Recommendations for the authors):*

This article presents a large number of interesting observations. However, as I see it, the current manuscript will be very difficult to read and appreciate for most readers. The main reason, to my point of view, is that the paper is very long and it is very difficult to keep in focus when reading it. I have some suggestions to try to improve this:

1) In general, and especially in the Results section, the results are presented with excessive technical details (e.g. the kinds of siRNA, the plugins or microscopes used for the imaging etc), which should be specified in all detail in the methods section, but not so much in the results. I think this massively streamlining the Results section accordingly would improve the readability of the manuscript.

2) There are different "sub-stories" within this manuscript. I would suggest the authors to put them in two main bodies of results and clearly separate them (one of the nice things about *eLife* is that the format in which you write your paper is pretty open and free, so the authors might find ways to be more clear here).

3) The discussion is a bit repetitive with the summary of all experimental results. I'd suggest that the discussion is merged with the results or shortened so it is a real Discussion section and not mainly a summary of the experimental results.

4) Particularly, the abstract is very (too) long. It should be streamlined and simplified, highlighting the most important findings. I'd ask the authors to write it in a more concise and logical manner, because now it reads as a summary of the results, which appear somehow disconnected to one another.

5) Lipid strips can provide first line of evidence for specific lipid binding, but at the moment it is not clear with PIP (if any) is preferred by SNX32. Could the authors do flotation experiments in which they incubate liposomes containing (or not) PI3P vs PI4P together with the SNX32 Δ C (which they have purified successfully and contains the PX domain)?

*Reviewer #3 (Recommendations for the authors):*

While the data in this paper is largely interesting, the presentation was extremely difficult to follow. It often left like the authors were trying to connect so many things and highlight so many points of emphasis the overall general role of SNX32 in intracellular trafficking difficult to understand. The test was highly repetitive and in every section of the manuscript I often was lost to what major points the authors were trying to make. It also feels like, there are two major stories here 1. Being the interaction partners of SNX32 and SNX32 roles in TfR and CIMPR trafficking and 2. The role of SNX32 in BSG trafficking and the role of this in neuronal differentiation. These two points could easily be two separate papers or major streamlining of this manuscript is needed to make it easier to understand the major points.

Is SNX32 role largely to direct trafficking steps from early endosomes to recycling endosomes, back to the plasma membrane? And is this a general mechanism for all recycled receptors or specific for TfR, CIMPR, and BSG trafficking?

[Editors' note: further revisions were suggested prior to acceptance, as described below.]

Thank you for resubmitting your work entitled "Insights into cargo sorting by SNX32 and its role in neurite outgrowth" for further consideration by *eLife*. Your revised article has been evaluated by Vivek Malhotra (Senior Editor) and a Reviewing Editor.

The manuscript has been improved but there are some remaining issues that need to be addressed, as outlined below:

1) Basically, the reviewers are mostly happy with the new version of the manuscript. I'd like to suggest that they try to take into account as much as possible the final recommendations from the reviewers before submitting a final version of the manuscript for acceptance.

*Reviewer #1 (Recommendations for the authors):*

I appreciate that the authors made a major effort to address comments that were raised in the review process. I do have, however, some remaining issues, listed below.

1. I do believe that manuscript could still benefit from reorganizing many of the figures, for example by putting consecutive panels next to each other (especially the main text Figures 3-5).

2. In Figure 3 it is quite confusing to compare panels E-F with panels L and Q (upper line) as they show similar things, namely the redistribution of CIMPR upon KD of SNX32. Perhaps authors should consider shifting some of these results to the supplementary data. It is also a bit confusing that some panels are labeled with αCIMPR and αEEA1 and others not (was antibody staining used only in some?), as well as that color assigned to CIMPR microscopy images is changed (especially in E-K).

---

## [Author Response]

Essential revisions:1)We all agree that this paper has the potential to become an important paper. However, in its present state, the manuscript is very difficult to follow. The authors should make an effort to rewrite and organize the paper in a manner that helps readers understand and appreciate the data The individual reviews make some suggestions, but the authors should make a substantial effort in improving/rewriting the manuscript, so it flows clearly and the readers are not lost in details.

We once again thank the reviewer for the detailed comments and suggestions, which helped in improving the overall presentation of the data reported in this manuscript. We sincerely apologise for the errors and omissions which happened during the preparation. We hope you will find the revised manuscript better.

Regarding the experiments, some simple experiments/clarifications will be required to strengthen the proposal (see individual reports for details):2) Explain/quantify better what you mean by "colocalization" in the fluorescence microscopy images.

We would like to thank the reviewer for the suggestion. The automated image analysis software, Motion Tracking^1, 2^ (http://motiontracking.mpi-cbg.de), which was used for the analysis, has an optimised algorithm for detecting the inhomogeneous background of fluorescent cytoplasm. The objects are defined by probabilistic threshold and as such, are detected as additional intensity on top of the background.

We identified each puncta as an object and further quantified their physical properties such as size, intensity, and area. For better clarification, we have shown in Author response image 1 a representative image with EEA1 objects depicted in magenta colour and CIMPR object depicted in green colour.

**Author response image 1. sa2fig1:** (**A**) HeLa cells stained with EEA1(magenta) and CIMPR (green), identified as objects by motion tracking and (**B**) their associated contour. (**C**) Inset is showing the magnified boxed region with one object. (**D**) The representative intensity distribution of this object as analysed by motion tracking. Further, the colocalisation in such a condition is calculated by analysing the total area overlap between the individual object.

As further depicted in Author response image 2 the total area of object A overlapping with total area of object B is less than 35% of total area of object A which is not enough to cross the minimum threshold of 35% to consider A colocalising to B. Whereas the total area of object B overlapping with object A is more than 30% of the total area of object B. So, in the following figure based on the analysis carried out by Motion tracking it will be considered that the object B is colocalising with object A whereas the vice versa is not true.

Additionally in the revised MS we have incorporated the colocalisation analysis carried out using ImageJ, another image analysis platform. The information is included in the text from line 705-735.

3) Add missing information in the methods section.

We apologise for the unintentional experimental detailing omissions, which are now incorporated into the text accordingly:

Super resolution sample preparation is now correctly labelled line No.: 776

4) Provide further evidence of PI3P vs PI4P binding (Reviewers #1 and #2 suggest some experiments along these lines).

We appreciate the reviewer suggestion for alternative approaches to show the lipid affinity of SNX32ΔC, which also helped us in improving the overall interpretation of the data. We now showed using the assay, referred to as proximity-based labelling of membrane-associated proteins (PLiMAP) ^3, 4^, as an additional evidence of lipid affinity of SNX32ΔC protein. The assay is based on UV activation of a fluorescent lipid reporter, which in turn crosslinks with proteins bound to membranes and renders them fluorescent. PLiMAP assays were carried out as described previously^3, 4^, with a few modifications. Proteins and liposomes were mixed at a final concentration of 2 and 200 µM, respectively in a total volume of 30 µl of 20 mM HEPES (pH 7.4) with 150 mM NaCl buffer. The reaction mix was transferred to a 96-well plate and incubated for 30 min at 25 ºC in the dark. The plate was then exposed to 365 nm UV light (UVP crosslinker CL^-^1000L) and an energy setting of 200 mJ.cm-2 for 1 min. The plate was placed at a distance of 3 cm from the lamp. Sample buffer was then added to the reaction mixture, aliquoted out into an eppendorf, boiled at 99 ºC, and resolved using SDS-PAGE. Gels were first scanned for BODIPY fluorescence on an AmershamTM TyphoonTM Bimolecular Imager (Cytiva Lifesciences) and later fixed and stained with Coomassie Brilliant Blue (Line No.: 844-863).

Proximity-based Labelling of Membrane Associated Proteins (PLiMAP) used the known PI(3)P binder SNX12^5, 6^ and PI(4)P binder 2XP4M domain of SidM^7^ were as controls. As shown in Figure S3 B, the SNX32 PX domain showed no binding to either PI(3)P or PI(4)P, indicating that its subcellular localisation to compartments rich in these lipids may not be through direct lipid-protein interaction. Alternatively, SNX32 may have much weaker binding affinities compared to the controls used here. The information is included in the text from line No.: 220-224; Figure 3—figure supplement 2 B. Accordingly, we have also modified our discussion.

5) Discuss if SNX32's main role is largely to direct trafficking steps from early endosomes to recycling endosomes, back to the plasma membrane? And is this a general mechanism for all recycled receptors or specific for TfR, CIMPR, and BSG trafficking? Also, discuss the possibility of cooperative binding with other proteins on endosomes.

Our results shows that the role of SNX32 is not just constrained to the trafficking from early endosome to recycling endosome, but also to the retrograde trafficking pathway as well (Line 557-570). In our study the SNX32 is involved in the transferrin recycling to the plasma membrane whereas in the case of CIMPR the SNX contributes to its trafficking from early endosome to the Golgi. However, our data could only suggest that the surface population of BSG is reduced upon the depletion of the SNX. Further investigations will be required to dissect that the detailed intracellular itinerary of BSG which is guided by SNX32. Moreover, the cooperative binding of the heterodimeric partners could also be crucial in the cargo recognition stage. It is noted in heterodimeric associations of SNX1/SNX5 and SNX1/SNX6, while SNX5, SNX6 lead cargo recognition^8^, SNX1/SNX2 are known to contribute to membrane remodelling establishing the mutual cooperativity of the individual SNX proteins during cargo sorting. Also, it has been reported in case of SNX3-retromer^7^, SNX27-SNX1/2^8^ the interaction of the complex improves the affinity with which the complex binds with the cargo. Thus, it could be hypothesised that the cooperative interactions of the heterodimeric complex facilitate the cargo recognition, membrane recruitment as well as membrane remodelling. Discussed in the manuscript file form line 537-544

Reviewer #1 (Recommendations for the authors):Conclusions of this paper are mostly well supported with experiments; however, some points need additional clarification. I have suggestions and comments that I believe should be addressed before publication.Comments:1. Authors state that expression of SNX32 in HeLa cells is comparable to U87MG and Neuro2a although graph in Figure S2A shows reduced by half expression in Neuro2a. This should be clarified in the text. Also, statistical analysis is missing in this graph.

We understand the concern raised by the reviewer; accordingly, we have modified the text. Our intention by showing the Figure 2—figure supplement 1 A, was to explain that the expression of SNX32 in HeLa cells was comparable with that in human brain-derived cells such as U87MG (Line No. 164-167), based on which we went ahead to do various biochemical and imaging-based investigations in HeLa cells. We also did the analysis and the plot (Figure 2—figure supplement 1 A) is now revised to incorporate the significance.

2. In the method description for the colocalization analysis authors state that >35% overlap is considered as colocalization, although in Figure 2B Rab11 or TGN46 are hardly colocalized with SNX32 (less than 20%). However, authors state in the description of the confocal experiment that colocalization was visible (line 187). Authors should rephrase this part to indicate that these results come rather from live-cell microscopy or super-resolution microscopy experiments.

The reviewers rightly pointed out the confusion created due to the presentation of the data, in the following section for a better understanding we have elaborated the detailed methodology of the colocalisation calculations used throughout the current study.

The colocalisation of SNX32 and Rab11 was indeed calculated with the confocal images using the automated image analysis software, Motion Tracking^1, 2^ (http://motiontracking.mpi-cbg.de), has an optimised algorithm for detecting the inhomogeneous background of fluorescent cytoplasm. The objects are defined by probabilistic threshold and as such, are detected as additional intensity on top of the background. Line No.:705-735.

We identified each puncta as an object and further quantified their physical properties such as size, intensity, and area. For better clarification, we have shown in author response image 1 a representative image with EEA1 objects depicted in magenta colour and CIMPR object depicted in green colour.

As further depicted in Author response image 2 the total area of object A overlapping with total area of object B is less than 35% of total area of object A which is not enough to cross the minimum threshold of 35% to consider A colocalising to B. Whereas the total area of object B is overlapping with object A is more than 35% of the total area of object B. So, in the following figure based on the analysis carried out by Motion tracking it is considered that the object B is colocalising with object A whereas the vice versa is not true.

Additionally in the revised manuscript we have incorporated the colocalisation analysis carried out using ImageJ, another image analysis platform which showed a Pearson-Corelation coefficient of ~24% between GFP-SNX32 and TGN46.

The live cell videos and SIM images were recorded as a supportive data for the low colocalisation observed in the confocal images. Further, we now provide GM130, a cis-golgi marker and GFP-SNX32’s colocalisation (2.2%). The analysis shows that in contrast to TGN46 (4.6%) and Golph3 both of which are trans-golgi markers, GFP-SNX32 show only a 2.2% colocalisation with GM130 a cis-golgi marker (Figure 2—figure supplement 1 B, Extended data 1). Which was confirmed using Olympus Spinning disk OSR super resolution imaging (Extended data 1 A-C). Line No.: 176-180.

As it could appreciated from the live cell video the interaction of Golgi and SNX32 is very transient (video. 3) and the super-resolution images (Figure 2F) shows that mCherrySNX32 punctae associated with Golgi scaffolds in a partial colocalisation manner which could be an explanation for the low colocalisation values as observed in confocal imaging.

3. Super-resolution microscopy videos (Supplementary Videos 2,3) seem hard to interpret as colocalization. Here (including Figure 2FG), some additional quantifications would be beneficial. Additionally, since there is quite a substantial difference in colocalization visible by confocal and super-resolution imaging in case of Golgi marker and Rab11, it requires a negative control with another organelle marker that does not show colocalization. Moreover, it is hard for the reader to refer it to the appropriate section in the methods section ("SIM sample preparation"), as the two are not linked by any key word (either "super-resolution" or SIM).

We sincerely thank the reviewers for rightly pointing out that the speed of the super resolution microscopy images was making them hard to interpret. The super resolution videos (video 2, video 3) are now slowed down so as to help the reader appreciate the observation of mCherry-SNX32 associated with GFP-Golph3 marked Golgi scaffold. Similarly in case of mCherry-Rab11 and GFP-SNX32 the punctate are not completely overlapping. In either case the localisation suggesting it to be of transient nature thus supporting the same shown in live cell imaging videos. The additional qualifications are now incorporated in which we utilised the Image J image processing platform to calculate the Pearson’s correlation coefficient that was determined after Costes’ automatic threshold by using ImageJ-Fiji between TGN46 and SNX32.which showed a colocalisation value of ~24%. Additionally, as per the reviewer suggestion we now included the colocalisation analysis between GM130 (a cis-golgi marker) and SNX32 which showed only 2.6% colocalisation. We further utilised the Olympus SpinSR microscope to acquire super resolution image of the same emphasising on the low colocalisation between the two proteins thus serving as a negative control. The methods section referring to the SIM sample preparation is now relabelled as Super-resolution sample preparation (line No.: 776, 731-735, 175-180).

4. Details of all qRT-PCR should be added to the methods sections. Primers used should be listed in the final version of the manuscript.

We sincerely apologise for the unintentional omission of the RT PCR experimental details, which are included in the text accordingly. The details of qPCR is now added in the methods sections. With a subheading “RNA extraction and quantitative real-time PCR (Line No.661-672)” describing the method “Total RNA was extracted from the cells using RNA easy kit (Qiagen, Cat. 74104), and cDNA was prepared using the High-Capacity RNA-to-cDNA kit (Life Technologies, Cat. 387406). Real-time qPCR reactions were performed using the SYBR Green Kit and corresponding primers on Applied Biosystems 7300 Real-Time PCR System or Thermo Quant Studio3.0.” The details of all the Primers, siRNA sequences and shRNA sequences used in study are now added in the Supplementary File 1.

5. Authors predict heterodimeric BAR structures of SNX32 with SNX1 and SNX4 with AlphaFold2-Multimer; however, details of these analyses are missing from the methods section. Input sequences (or at least residue-ranges) should be stated as well as details of how the obtained models were analyzed that ultimately led to identification of key residues.

We sincerely apologise for not providing sufficient information about the AlphaFold2-Multimer prediction in the methods section. The input sequence details are now incorporated along with the details of the analysis. The predictions of dimeric models were performed using AlphaFold2-Multimer (ColabFold)^9-11^. The amino acid sequences corresponding to the BAR domain of SNX1 (NCBI RefSeq: NM_003099.5), SNX4 (NCBI RefSeq: NM_003794.4), and SNX32 (NCBI RefSeq: NP_689973.2) with residue numbers 303-522, 205-448 and 180-400, respectively, were used as input. Top-ranked models with the best prediction quality (higher overall pLDDT score) were selected for further analysis. The obtained models were analysed for intermolecular polar interactions (hydrogen bonds and salt bridges) to determine potential key residues participating in the dimer formation. Analysis of the structures and generation of molecular graphics images was carried out in PyMol (version 2.5.2; Schrödinger). Buried surface area calculations of the dimeric interface were performed using the PDBsum webserver (Ref: PMID: 28875543). Line number :684-698.

6. The PIP-strip experiment (Figure 2H) does not show specificity for PI(3)P or PI(4)P, however authors make a point that it is PI(3)P and PI(4)P that PX of SNX32 binds to and directs to endosomes. This statement in the abstract is not fully justified by experiments. Even though, treatment with PI-inhibitors, shows re-localization from endosomes to the cytoplasm, it needs to be considered that specific subcellular localization could be mediated by interactions with other endosomal partners as well as cooperativity in membrane binding by PX and BAR domains (as suggested by different results with either deltaC or deltaN version of the protein and with very minor effect on the full length). Here, additional experiments looking at least at SNX1 and SNX4 localization upon inhibitor treatment are required as well as more extensive discussion of the results.

We certainly agree with the reviewer’s suggestion that the specific subcellular localisation could be mediated by interactions with other endosomal partners as well as cooperativity in membrane binding by PX and BAR domains. Indeed our recent experiments using PLiMAP^3, 4^/ liposome pelleting assay does not show a binding with PI(3)P, PI(4)P and therefore strengthens the hypothesis that the subcellular localisation of SNX32 requires other interacting proteins. And accordingly, the cooperativity in membrane binding of PX and BAR domain is a possibility. Accordingly, the Discussion section is now revised “Line number:518-529”.

As per the suggestion the effect of Wortmannin in the localisation of SNX1 and SNX4 was investigated and it was observed that the effect of Wortmannin was much pronounced in the localisation of GFP-SNX1 and GFP-SNX4 compared to SNX32, it is now incorporated in the results (Line 209-210) and the effect of inhibitors on localisation of SNX32 is further discussed “In HeLa cells, the results from the colocalisation studies on SNX32 in the absence or presence of SNX1/SNX4 (Figure 3 A, Figure 4 A, Figure 2—figure supplement 1 D-F) or the PI3K inhibitor-wortmannin (Figure 2 —figure supplement 1 K-M) indicate that the BAR domain-mediated interaction of the SNX with SNX1/SNX4 may contribute to its recruitment to EEA1, harbouring early endosomes. Similar interdependence was also observed for SNX6 and SNX27, where their hetero-dimerisation with SNX1/2 or SNX1, respectively, was shown to be important for membrane recruitment ^12, 13^. Though the contribution of heteromeric interactions in membrane localisation of SNX32 is in place, the role of PIns affinity of the PX domain cannot be disregarded. The results from our cellular localisation study on SNX32-PX in the presence or absence of Wortmannin (Figure 2—figure supplement 1L), indicated that SNX32’s PX domain might also contribute to its localisation on endosomes. Likewise, its affinity towards PI4P, as evident from its localisation in PAO-treated cells (Figure 2K), corroborates well with SNX32’s association with Golgi/recycling compartments. Although this was further supported by the results from the PIP strip-based studies (Figure 3—figure supplement 2 A) using His-SNX32ΔC, the PLiMAP investigations did not show any detectable binding of His-SNX32ΔC to PI(3)P or PI(4)P(Figure S3 B). However, the apparent contradiction of the above observations could be explained by SNX32’s low PIns binding affinities. Based on these observations we hypothesise that as reported earlier for SNX1^29^/SNX4^31^, both the PX domain as well as a the BAR domain contribute to the membrane recruitment of SNX32^14, 15^ (Line 506-529).

7. Authors show that SNX6 and SNX32 have overlapping roles in TfR trafficking; however, do not provide such analysis for CIMPR (for example in Figure S3L).

We appreciate that the reviewers found the overlapping role of SNX6 and SNX32 interesting and further suggested to investigate the same in CIMPR trafficking. As per the suggestion we have extended analysis for CIMPR, and added all the results in the revised manuscript (line No. 224-254). We have incorporated the results of our double knockdown of SNX32, SNX6 as well as SNX32, SNX1 in the steady state distribution of CIMPR. We observed that the SNX1 knockdown (KD)/ SNX6KD/ SNX32KD (Figure 3—figure supplement 1 A) resulted in the steady-state redistribution of CIMPR (Figure 3 E- H), which was also seen to be correspondingly more with EEA1 positive early endosomes (Figure 3 I). Moreover, the double KD of SNX1 and SNX32 did not show any significant difference in the redistribution of CIMPR (Figure 3 I-J). On the contrary the downregulation of SNX6 along with SNX32 showed a significant increase in the redistributed CIMPR (Figure 3 I-K) (Line No.225-255).

Reviewer #2 (Recommendations for the authors):This article presents a large number of interesting observations. However, as I see it, the current manuscript will be very difficult to read and appreciate for most readers. The main reason, to my point of view, is that the paper is very long and it is very difficult to keep in focus when reading it. I have some suggestions to try to improve this:1) In general, and especially in the Results section, the results are presented with excessive technical details (e.g. the kinds of siRNA, the plugins or microscopes used for the imaging etc), which should be specified in all detail in the methods section, but not so much in the results. I think this massively streamlining the Results section accordingly would improve the readability of the manuscript.

We appreciate the reviewer’s suggestion, and accordingly the extensive detailing is now removed from the Results section and included in figure legends or on Materials and methods as and when appropriate. Also, we have reorganised the manuscript including the Discussion section for the better readability. We sincerely hope the readability of the manuscript is improved with all the new observations and edits.

2) There are different "sub-stories" within this manuscript. I would suggest the authors to put them in two main bodies of results and clearly separate them (one of the nice things about eLife is that the format in which you write your paper is pretty open and free, so the authors might find ways to be more clear here).

We are thankful to the reviewer for appreciating the Sub-stories and suggesting the separation of the stories which is now incorporated in the Manuscript. The substory of SNX1-SNX32 trafficking CIMPR is now separated under the sub heading “SNX32 regulates CIMPR trafficking (225-255)” whereas the role of SNX32-SNX4 trafficking of Transferrin receptor is divided under the sub heading “SNX32 regulates transferrin trafficking (256-293)” and “interplay of SNX32 with SNX4 and Rab11 in transferrin trafficking (294-321)”.

3) The discussion is a bit repetitive with the summary of all experimental results. I'd suggest that the discussion is merged with the results or shortened so it is a real Discussion section and not mainly a summary of the experimental results.

We thank reviewer for the suggestion; accordingly, the discussion is now separated from the results, and the results are linked in continuation by stating the aim of the section. The Discussion section is kept separated from the results so that the all the substories (CIMPR, TfR, BSG) could be combined and discussed together. The experiments were carried out in such a way that in many instances the observations made in HeLa was exploited during the investigations in NeuroGlial cell line, the observations such as EEA1 localisation of SNX32, the lipid affinity-based surface localisation are few such connections. Further, we appreciate the reviewer suggestion about the redundancy in the Discussion section which is now removed.

4) Particularly, the abstract is very (too) long. It should be streamlined and simplified, highlighting the most important findings. I'd ask the authors to write it in a more concise and logical manner, because now it reads as a summary of the results, which appear somehow disconnected to one another.

We would like to thank the reviewers for rightly pointing out the abstract being lengthy, we have reduced the text in the abstract so reflect the main observations presented in the text. We sincerely hope that the reviewers find the revised abstract as much more improved and fitting to the context.

5) Lipid strips can provide first line of evidence for specific lipid binding, but at the moment it is not clear with PIP (if any) is preferred by SNX32. Could the authors do flotation experiments in which they incubate liposomes containing (or not) PI3P vs PI4P together with the SNX32 Δ C (which they have purified successfully and contains the PX domain)?

We sincerely thank the reviewer suggestion for incorporating alternative methodology to show the lipid affinity of SNX32ΔC. Accordingly, we now showed using the assay, referred to as proximity-based labelling of membrane-associated proteins (PLiMAP)^3, 4^, as an additional evidence of lipid affinity of SNX32ΔC protein. The assay is based on UV activation of a fluorescent lipid reporter, which in turn crosslinks with proteins bound to membranes and renders them fluorescent. PLiMAP assays were carried out as described previously^3, 4^, with a few modifications. Proteins and liposomes were mixed at a final concentration of 2 and 200 µM, respectively in a total volume of 30 µl of 20 mM HEPES (pH 7.4) with 150 mM NaCl buffer. The reaction mix was transferred to a 96-well plate and incubated for 30 min at 25 ºC in the dark. The plate was then exposed to 365 nm UV light (UVP crosslinker CL^-^1000L) and an energy setting of 200 mJ.cm-2 for 1 min. The plate was placed at a distance of 3 cm from the lamp. Sample buffer was then added to the reaction mixture, aliquoted out into an eppendorf, boiled at 99 ºC, and resolved using SDS-PAGE. Gels were first scanned for BODIPY fluorescence on an AmershamTM TyphoonTM Bimolecular Imager (Cytiva Lifesciences) and later fixed and stained with Coomassie Brilliant Blue.

Proximity-based Labelling of Membrane Associated Proteins (PLiMAP) used the known PI(3)P binder SNX12^5, 6^ and PI(4)P binder 2XP4M domain of SidM^7^ were as controls. As shown in Figure , the SNX32 PX domain showed no binding to either PI(3)P or PI(4)P, indicating that its subcellular localisation to compartments rich in these lipids may not be through direct lipid-protein interaction. Alternatively, SNX32 may have much weaker binding affinities compared to the controls used here. The information is included in the Figure 2—figure supplement 2 (B), text from line 220-224 and 844-863.

Reviewer #3 (Recommendations for the authors):While the data in this paper is largely interesting, the presentation was extremely difficult to follow. It often left like the authors were trying to connect so many things and highlight so many points of emphasis the overall general role of SNX32 in intracellular trafficking difficult to understand. The test was highly repetitive and in every section of the manuscript I often was lost to what major points the authors were trying to make. It also feels like, there are two major stories here 1. Being the interaction partners of SNX32 and SNX32 roles in TfR and CIMPR trafficking and 2. The role of SNX32 in BSG trafficking and the role of this in neuronal differentiation. These two points could easily be two separate papers or major streamlining of this manuscript is needed to make it easier to understand the major points.Is SNX32 role largely to direct trafficking steps from early endosomes to recycling endosomes, back to the plasma membrane? And is this a general mechanism for all recycled receptors or specific for TfR, CIMPR, and BSG trafficking?

Our results shows that the role of SNX32 is not just constrained to the trafficking from early endosome to recycling endosome, but also to the retrograde trafficking pathway as well (Line 557-570). In our study the SNX32 is involved in the transferrin recycling to the plasma membrane whereas in the case of CIMPR the SNX contributes to its trafficking from early endosome to the Golgi. However, our data could only suggest that the surface population of BSG is reduced upon the depletion of the SNX. Further investigations will be required to dissect that the detailed intracellular itinerary of BSG which is guided by SNX32. Moreover, the cooperative binding of the heterodimeric partners could also be crucial in the cargo recognition stage. Besides, it is noted in heterodimeric associations of SNX1/SNX5 and SNX1/SNX6, while SNX5, SNX6 lead cargo recognition^8, 20, 22, 26^, SNX1, SNX2 are known to contribute to membrane remodelling establishing the mutual cooperativity of the individual SNX proteins during cargo sorting. Also, it has been reported in the case of SNX3-retromer^27^, SNX27-SNX1/2 ^13^ that the interaction among the individual members of the above complexes enhances their affinity to the cargo. Thus, it could be possible that the cargo recognition, membrane recruitment as well as membrane remodelling by SNX32 could be facilitated by the cooperative mode of interactions. 530-544.

[Editors' note: further revisions were suggested prior to acceptance, as described below.]

Reviewer #1 (Recommendations for the authors):I appreciate that the authors made a major effort to address comments that were raised in the review process. I do have, however, some remaining issues, listed below.1. I do believe that manuscript could still benefit from reorganizing many of the figures, for example by putting consecutive panels next to each other (especially the main text Figures 3-5).

We agree with the reviewer, accordingly the figure 3-5 is now reorganized by keeping the consecutive panels next to each other.

2. In Figure 3 it is quite confusing to compare panels E-F with panels L and Q (upper line) as they show similar things, namely the redistribution of CIMPR upon KD of SNX32. Perhaps authors should consider shifting some of these results to the supplementary data. It is also a bit confusing that some panels are labeled with αCIMPR and αEEA1 and others not (was antibody staining used only in some?), as well as that color assigned to CIMPR microscopy images is changed (especially in E-K).

We thank the reviewer for rightly pointing out the confusion created by side-by-side representation of multiple panels showing similar results. We have now moved some of the supporting results to supplementary files as detailed below and the manuscript text is edited accordingly.

Figure 3:

Figure 3L-M (shRNA related results on CIMPR trafficking) are now moved to Figure3—figure supplement 1 C-F.

Figure 3 N-O (CD8-CIMPR related results on CD8 uptake assay) are now moved to Figure3—figure supplement 1 I-K

Figure 4:

Figure 4 J-K (shRNA related data on transferrin uptake) is now moved to Figure 4 —figure supplement 1 A-B

We apologise for creating confusion in the labelling and pseudo colour assigned to CIMPR and EEA1 which is now corrected in Figure 3, its supplement, and also cross verified in all other panels.

In addition, we have also corrected the label on Figure 7—figure supplement 1 which was by mistake labelled as ‘C-E’ in the figure panels and ‘A-C’ in the legends as well as in the previous version of the manuscript. In the re-revised manuscript, it is ‘A-C’ in the panels, legends and the manuscript text.

References

1. Kalaidzidis, Y. Intracellular objects tracking. European journal of cell biology 86, 569-578 (2007).

2. Rink, J., Ghigo, E., Kalaidzidis, Y. & Zerial, M. Rab conversion as a mechanism of progression from early to late endosomes. Cell 122, 735-749 (2005).

3. Jose, G.P. & Pucadyil, T.J. PLiMAP: Proximity‐Based Labeling of Membrane‐Associated Proteins. Current Protocols in Protein Science 101, e110 (2020).

4. Jose, G.P., Gopan, S., Bhattacharyya, S. & Pucadyil, T.J. A facile, sensitive and quantitative membrane‐binding assay for proteins. Traffic 21, 297-305 (2020).

5. Priya, A. et al. Essential and selective role of SNX12 in transport of endocytic and retrograde cargo. Journal of cell science 130, 2707-2721 (2017).

6. Pons, V. et al. SNX12 role in endosome membrane transport. PloS one 7, e38949 (2012).

7. Brombacher, E. et al. Rab1 guanine nucleotide exchange factor SidM is a major phosphatidylinositol 4-phosphate-binding effector protein of Legionella pneumophila. Journal of biological chemistry 284, 4846-4856 (2009).

8. Simonetti, B. et al. Molecular identification of a BAR domain-containing coat complex for endosomal recycling of transmembrane proteins. Nature cell biology 21, 1219-1233 (2019).

9. Jumper, J. et al. Highly accurate protein structure prediction with AlphaFold. Nature 596, 583-589 (2021).

10. Evans, R. et al. Protein complex prediction with AlphaFold-Multimer. BioRxiv (2021).

11. Mirdita, M. et al. ColabFold-Making protein folding accessible to all. (2021).

12. Hong, Z. et al. The retromer component SNX6 interacts with dynactin p150 Glued and mediates endosome-to-TGN transport. Cell research 19, 1334-1349 (2009).

13. Yong, X. et al. SNX27-FERM-SNX1 complex structure rationalizes divergent trafficking pathways by SNX17 and SNX27. Proceedings of the National Academy of Sciences 118 (2021).

14. Traer, C.J. et al. SNX4 coordinates endosomal sorting of TfnR with dynein-mediated transport into the endocytic recycling compartment. Nature cell biology 9, 1370-1380 (2007).

15. Carlton, J. et al. Sorting nexin-1 mediates tubular endosome-to-TGN transport through coincidence sensing of high-curvature membranes and 3-phosphoinositides. Current biology 14, 1791-1800 (2004).

16. Fridy, P.C. et al. A robust pipeline for rapid production of versatile nanobody repertoires. Nature Methods 11, 1253-1260 (2014).

17. Kubala, M.H., Kovtun, O., Alexandrov, K. & Collins, B.M. Structural and thermodynamic analysis of the GFP: GFP‐nanobody complex. Protein Science 19, 2389-2401 (2010).

18. Johnson, J.L. et al. Munc13-4 is a Rab11-binding protein that regulates Rab11-positive vesicle trafficking and docking at the plasma membrane. Journal of Biological Chemistry 291, 3423-3438 (2016).

19. Longatti, A. et al. TBC1D14 regulates autophagosome formation via Rab11-and ULK1-positive recycling endosomes. Journal of Cell Biology 197, 659-675 (2012).

20. Simonetti, B., Danson, C.M., Heesom, K.J. & Cullen, P.J. Sequence-dependent cargo recognition by SNX-BARs mediates retromer-independent transport of CI-MPR. Journal of Cell Biology 216, 3695-3712 (2017).

21. Kvainickas, A. et al. Cargo-selective SNX-BAR proteins mediate retromer trimer independent retrograde transport. Journal of Cell Biology 216, 3677-3693 (2017).

22. Niu, Y. et al. PtdIns (4) P regulates retromer–motor interaction to facilitate dynein–cargo dissociation at the trans-Golgi network. Nature cell biology 15, 417-429 (2013).

23. Van Weering, J.R. et al. Molecular basis for SNX‐BAR‐mediated assembly of distinct endosomal sorting tubules. The EMBO journal 31, 4466-4480 (2012).

24. Wassmer, T. et al. A loss-of-function screen reveals SNX5 and SNX6 as potential components of the mammalian retromer. Journal of cell science 120, 45-54 (2007).

25. Lord, S.J., Velle, K.B., Mullins, R.D. & Fritz-Laylin, L.K. SuperPlots: Communicating reproducibility and variability in cell biology. Journal of Cell Biology 219 (2020).

26. Seaman , M.N.J. Cargo-selective endosomal sorting for retrieval to the Golgi requires retromer. Journal of Cell Biology 165, 111-122 (2004).

27. Lucas, M. et al. Structural mechanism for cargo recognition by the retromer complex. Cell 167, 1623-1635. e1614 (2016).